# PLANNER: Generating Diversified Paragraph via Latent Language Diffusion Model

**Yizhe Zhang, Jiatao Gu, Zhuofeng Wu, Shuangfei Zhai, Josh Susskind, Navdeep Jaitly**
Apple
`{yizzhang, jgu32, zhuofeng_wu, szhai, jsusskind, njaitly}@apple.com`

## Abstract

Autoregressive models for text sometimes generate repetitive and low-quality output because errors accumulate during the steps of generation. This issue is often attributed to exposure bias – the difference between how a model is trained, and how it is used during inference. Denoising diffusion models provide an alternative approach in which a model can revisit and revise its output. However, they can be computationally expensive and prior efforts on text have led to models that produce less fluent output compared to autoregressive models, especially for longer text and paragraphs. In this paper, we propose PLANNER, a model that combines latent semantic diffusion with autoregressive generation, to generate fluent text while exercising global control over paragraphs. The model achieves this by combining an autoregressive "decoding" module with a "planning" module that uses latent diffusion to generate semantic paragraph embeddings in a coarse-to-fine manner. The proposed method is evaluated on various conditional generation tasks, and results on semantic generation, text completion and summarization show its effectiveness in generating high-quality long-form text in an efficient manner.

## 1 Introduction

| Prefix: Horrible hotel. Horrible hotel. Seriousy, horrible hotel. | | 1-gram | 2-gram | 3-gram |
|---|---|---|---|---|
| **FT (Greedy decoding):** The hotel was a nightmare. The rooms were filthy, the bathrooms were filthy, the food was disgusting, the food was disgusting. The food was disgusting. The food was disgusting. The food was disgusting. The food was disgusting. The food was disgusting…. | | The: 100.00% | The hotel: 100.00% | The hotel was: 100.00% |
| **FT (Top-p sampling):** Terrible, awful hotel. Horrible, awful hotel. I'm going to the awful hotel. I'll never get in. What the hell is this hotel? It's awful. I don't like it. Where? It's awful. I think they have a bathroom there. I'm going to the awful hotel. I'll never get in. I want out of here. This hotel is horrible. I'm going to the awful hotel. I'll never get in. I want out of here… | | I: 13.39%
Horrible: 12.20%
The: 8.27%
Terrible: 5.91%
This: 4.33% | Horrible hotel: 12.50%
Terrible hotel: 6.05%
This is: 2.82%
I hate: 1.61%
I mean: 1.61% | Horrible hotel. Horrible: 3.48%
Horrible hotel. I: 2.61%
Terrible hotel. Terrible: 2.17%,
Terrible hotel. Horrible: 1.30%
I could not: 0.87% |
| **Ours (Greedy decoding):** Staff was unhelpful. So rude from the front desk. The hotel tried to lose the reservation of my family which I had with the hotel. This hotel is way overpriced. Hallways smelled bad. Carpet was disgusting. I'll never go back. I don't recommend the hotel. I can stay anywhere else!! | | I: 10.55%
The: 5.08%
We: 3.52%
Hotel: 2.73%
They: 2.73% | I was: 1.56%
Had to: 1.56%
The room: 1.17%
Dirty and: 1.17%
I called: 1.17% | The room was: 1.18%
The walls were: 0.78%
I was told: 0.78%
I called the: 0.78%
I booked through: 0.78% |

Figure 1: Left: With a repetitive prompt, the finetuned GPT-2 large model (774M, **FT**) is still attracted to self-reinforced repetition (highlighted text) even under top-p sampling (K=50, p=0.92). Right: the most frequent first n-grams of the generations for each method, derived from 512 generation roll-outs. Our proposed method results in a more diversified generation robust to the ill-composed prompt.

Autoregressive models trained with a teacher forcing strategy (Williams & Zipser, 1989) are considered the gold standard for text generation. However, a significant drawback of this approach is that it lacks the ability to correct any mistakes made during the generation process which can lead to errors that accumulate as the generation progresses. Previous work (Ott et al., 2018; Holtzman et al., 2019; Welleck et al., 2019; Xu et al., 2022) has observed that *deterministic decoding* methods have a

37th Conference on Neural Information Processing Systems (NeurIPS 2023).

tendency to generate consecutive repetitions at the word, phrase and sentence levels. For example, with repetitive prompt, the model can enter an *absorbing state* where it produces repetitive outputs with higher and higher confidence by *self-reinforcing* the pattern (Xu et al., 2022) (Fig. 1). Through our experiments, we have observed that such degeneration is more prevalent in *open-ended* tasks that allow the model greater freedom for creativity. Even for large language models, the generation can drift away from the desired semantics, especially when the model is poorly prompted or has high initial probabilities (Xu et al., 2022).

Why does using maximum likelihood decoding lead to repetitions during generation which is significantly different from the training data distribution? One possible explanation for this is "exposure bias" (Bengio et al., 2015) arising from the discrepancy between the training and inference phases in the teacher forcing training strategy. During training phase, the model focuses only on predicting the next token. However, during inference, predicting the next token alone can be myopic because the model may not have enough foresight to anticipate its impact on future generation. This can also be seen as the "distribution shift" issue of behavior cloning (de Haan et al., 2019), where the model is trained to mimic the expert's actions on the states encountered by the expert in the training data. However, small differences between the model and the expert can compound over multiple steps, leading the model to states it never encountered during training, rendering unreliable and undesirable predictions.

Although many approaches have been proposed to address this issue, such as adversarial models (Yu et al., 2017; Lamb et al., 2016; Zhang et al., 2017), reinforcement learning (Li et al., 2016b) or repetition penalties (Xu et al., 2022), they attempt to improve the *global* aspects of the generation by making *local* adjustments which still follow the autoregressive generation recipe. Diffusion models provide an alternative solution – the model can revisit and revise its output iteratively, potentially rendering more global control of the generation in a non-autoregressive manner. However, these text diffusion models can generate less fluent text compared to autoregressive ones (Gong et al., 2023). Also, when generating long text, the diffusion process involves multiple passes of the underlying denoising model over a long generation length, making it computationally expensive. The discrete nature of text also presents a challenge for diffusion models, which can suffer from "rounding errors" when converting between the text token and its embedding (Li et al., 2022; Lin et al., 2022).

Instead of performing diffusion on the original text or the corresponding word embeddings, we propose to apply diffusion techniques to the latent semantic space (Rombach et al., 2022; Lovelace et al., 2022). To achieve this, we learn a fixed number of continuous semantic tokens that encode salient information at the paragraph level. These tokens can then be used to reconstruct the original text. The latent diffusion can be additionally conditioned on an external signal to generate the semantic tokens. Finally, a decoder maps the obtained semantic tokens back to the raw text space. This process combines a non-autoregressive semantic diffusion approach with an autoregressive decoding technique. The semantic diffusion process handles the "planning", enabling the modification of semantics in a coarse-to-fine manner, while the decoder handles the "decoding" by translating the semantics into raw text, with less flexibility in controlling the meaning. We call our proposed method PLANNER (**P**aragraph-leve**L** Diffusio**N** model for **E**mbedding **R**epresentation).

Our contributions include: $(i)$ We propose a latent semantic diffusion model for paragraphs that incorporates both non-autoregressive semantic diffusion and autoregressive generation. This allows us to generate fluent text while being able to exercise global control inherited from a diffusion model. $(ii)$ We study the essential requirements for a good latent space for paragraph diffusion models. $(iii)$ We evaluate the effectiveness of our proposed method on various conditional generation tasks. Thanks to the iterative refinement of desnoising diffusion, our method enjoys less repetitive and more diverse generation, while maintaining good fluency and relevancy, comparing with autoregressive and text diffusion baselines (Li et al., 2022; Lin et al., 2022).

## 2 Preliminary

**Diffusion Probabilistic Models**  The standard diffusion model (DM) (Ho et al., 2020; Song & Ermon, 2019) learns the data distribution $p(x)$ by gradually denoising a normally distributed variable in a Markov chain of length $T$. The diffusion process can be viewed as a continuous-time stochastic process (Song et al., 2021b; Kingma et al., 2021) where the initial data point $\boldsymbol{x} \in \mathbb{R}^N$ is progressively corrupted by noise according to a predefined signal-noise schedule $\{\alpha_t, \sigma_t\}$, resulting in time-

dependent corrupted data $\{\boldsymbol{x}_t | t \in [0,1], \boldsymbol{x}_0 = \boldsymbol{x}\}$. The transition distribution is given by:

$$q(\boldsymbol{x}_t | \boldsymbol{x}_s) = \mathcal{N}(\boldsymbol{x}_t; \alpha_{t|s}\boldsymbol{x}_s, \sigma_{t|s}^2 I), \quad (1)$$

where $\alpha_{t|s} = \alpha_t / \alpha_s, \sigma_{t|s}^2 = \sigma_t^2 - \alpha_{t|s}^2 \sigma_s^2$, and $s < t$. When $\boldsymbol{x}_s = \boldsymbol{x}$, the marginal distribution $q(\boldsymbol{x}_t | \boldsymbol{x})$ is given as $q(\boldsymbol{x}_t | \boldsymbol{x}) = \mathcal{N}(\boldsymbol{x}_t; \alpha_t \boldsymbol{x}, \sigma_t^2 I)$. The diffusion model relies on a parametric function $\theta$ optimized to reverse the diffusion process by denoising $\boldsymbol{x}_t$ to the clean input $\boldsymbol{x}$. The model is trained using a weighted reconstruction loss:

$$\mathcal{L}(\theta) = \mathbb{E}_{\boldsymbol{x}_t \sim q(\boldsymbol{x}_t | \boldsymbol{x}), t \sim [0,1]} \left[ \omega_t \cdot \| \boldsymbol{F}_\theta(\boldsymbol{x}_t, t) - \boldsymbol{x} \|_2^2 \right], \quad (2)$$

where $\omega_t = \alpha_t^2 / \sigma_t^2, (s.t. \ \alpha_t^2 + \sigma_t^2 = 1)$ is the signal-to-noise-ratio (SNR) and $\boldsymbol{F}_\theta(\cdot)$ denotes the backbone denoising function. Sampling from the learned model can be performed using either ancestral sampling (DDPM) (Ho et al., 2020) or a deterministic DDIM sampler (Song et al., 2021a). While the DM is capable of generating high-quality samples, the fact that the corrupted data $\boldsymbol{x}_t$ shares the same space as the input $\boldsymbol{x}$ results in inefficient training (Jing et al., 2022) and difficulty in learning abstract and semantically meaningful latent spaces (Preechakul et al., 2022).

**Latent Diffusion Models**    To improve the efficiency, the Latent Diffusion Model (LDM) (Rombach et al., 2022) introduces an explicit separation between the compressive and generative learning phases of training diffusion models. It employs an autoencoding model consisting of an encoder $\mathcal{E}(\cdot)$ and a decoder $\mathcal{D}(\cdot)$ to learn a low-dimensional latent space that is perceptually equivalent to the image space when decoded, but with reduced computational complexity, while retaining the perceptual quality of generated samples. The reweighted objective for training LDM is given by:

$$\mathcal{L}(\theta) = \mathbb{E}_{\boldsymbol{z}_t \sim q(\boldsymbol{z}_t | \boldsymbol{z}), \boldsymbol{z} = \mathcal{E}(\boldsymbol{x}), t \sim [0,1]} \left[ \omega_t \cdot \| \boldsymbol{F}_\theta(\boldsymbol{z}_t, t) - \boldsymbol{z} \|_2^2 \right], \quad (3)$$

where $\boldsymbol{z}$ is obtained from $\mathcal{E}$ during training. The generated $\boldsymbol{z}$ can be decoded to image using $\mathcal{D}$.

## 3    Related Work

**Text diffusion models**    Early attempts on using diffusion models for discrete data used a noising processes which masked or randomly mutated the discrete tokens (Austin et al., 2021; Hoogeboom et al., 2021). Recently, Diff-LM (Li et al., 2022) and DiffuSeq (Gong et al., 2023) have instead used a continuous token embedding space, converting the continuous token embeddings to text via "rounding". Analog Bits (Chen et al., 2022) converts raw text into a set of bits and models them as analog bits with a continuous diffusion model. (Lovelace et al., 2022) performed diffusion model on the contextualized BART embeddings rather than on the word embedding space. (Zhu et al., 2022) has applied text diffusion to image-captioning and achieved good performance.

However, existing text diffusion models present several issues: *(i)* The varying length of the input text necessitates the prediction of additional length or superfluous paddings, and *(ii)* token generation in parallel may result in disfluent text and/or frequent repetitions especially when the generation is long. We instead employ the diffusion model to learn paragraph embeddings that contain fewer fixed-sized tokens, which allows for computational benefits and improved fluency.

**Text Variational Autoencoders**    Text VAEs (Bowman et al., 2016; Kim et al., 2018; Li et al., 2020) have been particularly useful for learning a smooth and interpretable representation space, as well as for generating diverse text. However, one of the challenges is the KL vanishing problem (Bowman et al., 2016), which results in the decoder disregarding the latent code sampled from the prior distribution during the inference stage. Our approach can be perceived as to address this issue by leveraging a more flexible prior distribution to ensure the codes can strongly influence the output text distribution.

## 4    PLANNER: A Language Diffusion Model on Paragraph Embeddings

We use latent diffusion to improve the diversity and fluency of paragraphs generated from the model. Our model comprises two parts (Fig. 2) - a paragraph embedder via variational autoencoder (VAE) that learns a meaningful and smooth latent space that corresponds to the original text space, and a diffusion model that generates latent codes corresponding to the semantics of longer text.

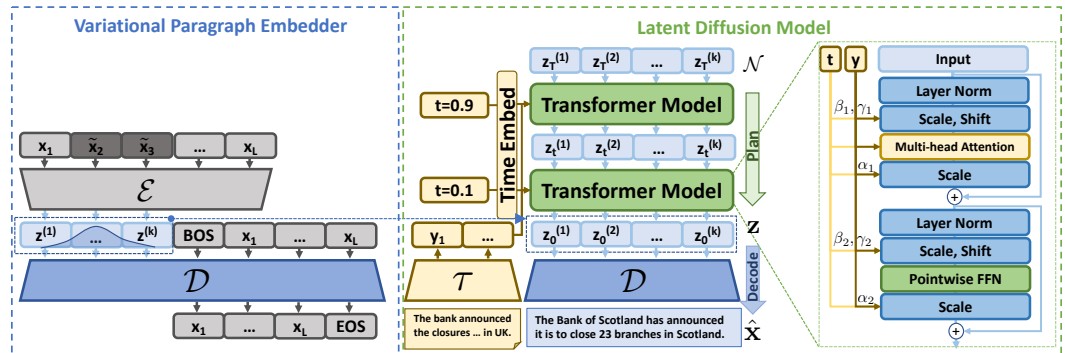

Figure 2: Model overview. Left: a variational paragraph embedder is learned to encode paragraph into a fixed amount of latent codes. Right: the latent diffusion model based on transformer block is applied to generate the latent codes. The decoder finally translates them into the text. (*BOS*: Begin of Sentence token, *EOS*: End of Sentence token)

## 4.1 Learning a Variational Paragraph Embedder

Instead of applying diffusion to tokens directly to generate long text, we propose to learn a set of latent codes $\boldsymbol{z} = \{z^{(1)}, \cdots, z^{(k)}\} \in \mathbb{R}^{k \times h}$, which we call *paragraph embeddings*, that capture the semantics in the target text (of length up to 512 tokens), where $h$ denotes the embedding dimension. These paragraph embeddings have shorter length, such as $k = 16$, than the original text.

To obtain such embeddings $\boldsymbol{z}$, we train a transformer-based encoder-decoder model. The architecture used for the autoencoder is shown in Fig. 2. The encoder $\mathcal{E}$ and decoder $\mathcal{D}$ construct a bidirectional mapping between the discrete data space and the latent code space. The paragraph embeddings $\boldsymbol{z}$ are extracted by taking the first $k$ hidden state vectors of dimension $h$ from the final layer of $\mathcal{E}$, which are fed into the initial steps of the decoder which is trained to reconstruct the original text. It's worth noting that the paragraph embeddings share the same hidden dimension $h$ as the word embeddings, and forming a manifold in the word embedding space. Pretrained BERT and GPT-2 models are used to initialize $\mathcal{E}$ and $\mathcal{D}$, respectively. The manifold of the learned embeddings ideally possesses several desirable properties, including low *conversion error*, *local smoothness* and *distributional smoothness*.

**Conversion error** Ideally, the original input $\boldsymbol{x}$ can be perfectly reconstructed via $\hat{\boldsymbol{x}} = \mathcal{D}(\boldsymbol{z}), \boldsymbol{z} = \mathcal{E}(\boldsymbol{x})$, and modeling the lower-dimensional continuous space $p(\boldsymbol{z})$ is equivalent to modeling $p(\boldsymbol{x})$. However, in practice a loss of information can occur when converting raw text into paragraph embeddings or when doing the reverse. We assess the conversion loss by computing the BLEU score (**BLEU_clean**) between the input $\boldsymbol{x}$ and the reconstruction $\hat{\boldsymbol{x}}$.

**Local smoothness** To generate target text that is fluent and consistent with the corresponding paragraph embeddings, it is essential to achieve a certain level of local smoothness in the paragraph embeddings space. Ideally, a slight variation in the input vector $\boldsymbol{x}$ would not cause a significant change in the resulting encoded vector $\mathcal{E}(\boldsymbol{x})$. Similarly, a small perturbation in the latent vector $\boldsymbol{z}$ should not lead to a significant change in the decoded vector $\mathcal{D}(\boldsymbol{z})$. Otherwise, the error accumulated in the diffusion process when generating $\boldsymbol{z}$ could result in an inaccurate realization of the desired semantics. To accomplish this, the denoising autoencoder is trained by substituting (Sub) input tokens with random tokens with probability $p$. The local smoothness is measured using the BLEU score (**BLEU_robust**) between the input $\boldsymbol{x}$ and the denoising output from corrupted input $\mathcal{D}(\mathcal{E}(\tilde{\boldsymbol{x}}))$, where $\tilde{\boldsymbol{x}} = \text{Sub}(\boldsymbol{x}, p = 0.3)$. The level of injected noise will affect both the conversion error and the local smoothness, and it is important to strike a balance between the two.

**Distributional Smoothness** The diffusion model may face difficulty in learning a distribution, $p(\boldsymbol{z})$, that is highly multimodal, or has density that are associated with a large Lipchitz constant (*i.e.*, has abrupt changes). Therefore, we employ a text VAE (Bowman et al., 2016; Li et al., 2020) to encourage the posterior to take on a form close to a Gaussian distribution. Specifically, we parameterize $q(\boldsymbol{z}|\boldsymbol{x})$ to be $\mathcal{N}(\mathcal{E}_\mu, \mathcal{E}_\nu)$ and maximize the objective $\mathcal{L}(\mathcal{E}, \mathcal{D}; \boldsymbol{x}) = \mathbb{E}_{q(\boldsymbol{z}|\boldsymbol{x})}[\log p(\boldsymbol{x}|\boldsymbol{z})] - \beta \cdot \text{KL}(q(\boldsymbol{z}|\boldsymbol{x}) \| p(\boldsymbol{z}))$. Here $\mathcal{E}_\mu$ and $\mathcal{E}_\nu$ represent the posterior mean and variance predictions of the encoder $\mathcal{E}$, while the

hyperparameter $\beta$ controls the strength of regularization. It is typically set to a small value to alleviate the notorious posterior collapsing issue (Bowman et al., 2016) in text VAE. To gauge the distributional smoothness of the paragraph embedding space, we select two examples, $x$ and $x'$ at random from the training set and interpolate their embeddings to compute $z_{\text{INT}} = \frac{1}{2}\mathcal{E}(x) + \frac{1}{2}\mathcal{E}(x')$. We then evaluate the perplexity ($\text{PPL}_{\text{int}}$) of the decoded interpolation $\mathcal{D}(z_{\text{INT}})$ using a GPT-2 model.

### 4.2 Planning then Decoding: A Latent Diffusion Model for Paragraph Embeddings

**Training phase** We now use the learned mean paragraph embeddings $z = \mathcal{E}_\mu(x)$ to train a continuous-time latent diffusion model as in Fig. 2 while keeping $\mathcal{E}$ and $\mathcal{D}$ frozen. We conducted experiments using two types of conditioning signal: $(i)$ class labels, such as positive or negative sentiment labels, and $(ii)$ raw text, such as preceding context or the document to be summarized. For class labels, we learned a label embedding $y \in \mathbb{R}^h$ to represent each class. For the raw text, we applied a conditional feature encoder $\tau$ to the input and used the hidden states at the last layer as $y \in \mathbb{R}^{c \times h}$, where $c$ represents the number of feature embeddings.

During training, we gradually add noise to $z$ via a cosine scheduler (Ho et al., 2020), and use a signal prediction scheme as the training objective (Kingma et al., 2021). For our denoising backbone model $F(\cdot)$, we use a transformer block similar to the one in the DiT (Peebles & Xie, 2022) model. Specifically, we fed $y$ and the time embedding $t \in \mathbb{R}^h$ into the model through two channels, namely cross-attention and adaptive layer norm (adaLN) (Peebles & Xie, 2022). For the cross-attention, the conditional embeddings $t$ and $y$ are concatenated into a sequence of length $c + 1$. The transformer block is modified to enable multi-head cross-attention to the conditional embeddings.

For the adaLN, we flattened and projected $y$ to a vector of $\mathbb{R}^h$ using a linear projection layer. We then added the projected $y$ to the time embedding $t$. Instead of directly learning dimension-wise scale and shift parameters ($\beta$ and $\gamma$) in the standard Layer Norm (LN), these parameters are regressed from the sum of the embeddings. In addition, dimension-wise scaling parameters $\alpha$ are regressed and applied immediately before any residual connections within the transformer block. This has been shown to be efficient and effective in image diffusion models (Peebles & Xie, 2022).

**Inference phase** During the inference process, we start with random Gaussian embeddings and use a fixed number of steps $T$ to generate the final $z$. The resulting embeddings are then used as inputs for $\mathcal{D}$ to generate the text using a *deterministic decoding* method like greedy decoding [1]. We provide discussion and ablation study on using stochastic decoding in the App. C

We applied the classifier-free guidance (CFG) (Ho & Salimans, 2021) during the inference steps. After each step, we apply a *dynamic thresholding* technique that was introduced in Imagen (Saharia et al., 2022) for post-processing. However, we do not use the rescaling step in Imagen because rescaling step can completely alter the underlying semantics, as we have not imposed any constraints to ensure that the generated output remains the same after rescaling the latent code. In contrast, for Imagen, where the generation takes place in the raw pixel space, rescaling will predominantly retain the shape information while altering only the contrast and brightness.

## 5 Experimental Setups

We tested the effectiveness of our model in three different conditional generation tasks including sentiment-guided generation, long-form text completion, and summarization. The tasks can require generating text of hundreds of tokens in length, making them suitable to assess model performance.

**Datasets** For the Sentiment-guided generation task, we used the TripAdvisor dataset provided by (Li et al., 2014). By exclusively selecting reviews with a rating of 1 or 5 and balancing the two ratings via subsampling, we acquired 218,509 reviews. For the text completion task, our model was assessed on two datasets: 1) the aforementioned TripAdvisor review dataset with postprocessing to remove reviews that are less than 20 or more than 512 tokens, result in 690,862 samples, and 2) one-tenth of the overall C4 datasets (Raffel et al., 2020), which contains 36.5M samples. For each sample, we extracted the initial two sentences from a paragraph as source context, and predicted the remainder of the text as target. The datasets were partitioned into training, validation, and test in the ratios of

---

[1]The aim of $\mathcal{D}$ is to accurately convert the $z$ into a meaningful text, thus deterministic decoding is desirable.

$(0.96, 0.02, 0.02)$. For the summarization task, we use CNN/DailyMail (Hermann et al., 2015) and XSum (Narayan et al., 2018).

**Automatic Evaluation Metrics** Following previous work (Gong et al., 2023), we assess the **fluency** of the generation by computing the perplexity (**PPL**) under a GPT-2 large model. We use **Ent-n** (Zhang et al., 2018) and **DIST-n** (Li et al., 2016a) and self-BLEU (**S-BL**) (Zhu et al., 2018) to evaluate lexical diversity. We present DIST-n and Ent-n metrics solely at $n = 1$ owing to their strong correlation of the varying $n$ values. We use **REP-n** to assess the extent of repetition in the generation following previous work (Welleck et al., 2019; Xu et al., 2022). For relevancy we use standard metrics following (Gong et al., 2023), including SacreBLEU (**BL**) (Post, 2018), ROUGE-L (**R-L**) (Lin, 2004) and BERTScore (**Score**) (Zhang et al., 2019). Details are provided in App. F.

**AuBLEU: Evaluating Denoising Capability** Our proposed model is a latent diffusion model, which differs from text diffusion models that operate directly on text or text embedding space. To comparing the denoising ability across different text diffusion models, we introduce a novel metric, named AuBLEU (**AuBL**). To compute the AuBLEU score, we first add varying levels of noise to each input text $\boldsymbol{x}$ by performing diffusion at $T$ different time steps $t_0 < t_1 < \cdots < t_T$, corresponding to a series of SNR $\omega_{t_0} > \cdots > \omega_{t_T}$. Next, we pass each corrupted input under different $\omega$ to the denoising backbone model and obtain the predicted output $\hat{\boldsymbol{x}}_i = \boldsymbol{F}(\boldsymbol{x}_{t_i})$. We then compute the BLEU score between each $(\hat{\boldsymbol{x}}_i, \boldsymbol{x})$ pairs and plot a curve with the x-axis representing $\alpha^2 = \frac{\omega}{1+\omega}$, where $\alpha^2$ is monotonically increasing with $\omega$ and ranges from $(0, 1)$, and the y-axis indicating the corresponding BLEU score. Finally, we compute the area under curve to obtain the AuBLEU score (see App. D for more details).

**Model Setups** We used the BERT-large and GPT-medium models as initialization for the encoder $\mathcal{E}$ and decoder $\mathcal{D}$ respectively. The embedding dimension $h$ was 1024, and the number of paragraph embeddings $k$ was set to 16, as increasing the number did not result in significant improvement in performance. We provide more analysis on the impact of $k$ in App. A.2 The learning rate was set to $2e-4$, and $\beta$ was set to $5e-6$. For the latent diffusion model, the channel size was set to 1024 to match the embedding dimension $h$, and the number of heads was set to 16 with 28 transformer layers. The total size of the latent diffusion model was 533M. The feature encoder $\tau$ was also jointly learned, and was initialized with a T5-large encoder. We use DDIM throughout our experiments as it shows better performance than DDPM. In all our experiments, we use 30 diffusion steps to generate the final $\boldsymbol{z}$, which strikes a good balance among the efficiency, diversity and relevance. In comparison, Diff-LM (Li et al., 2022) and Genie (Lin et al., 2022) report to use 200 steps and 2000 steps respectively to generate high-quality text. We set the CFG weights to be 2 and 5 for text completion and summarization tasks, respectively, based on generation performance on validation set. For summarization task, we also incorporate a shift noise scheduler based on (Hoogeboom et al., 2023). More details, including ablations on DDPM, number of diffusion steps and noise scheduler, are provided in App. F.

**Baselines** We compare our method with several baseline methods trained under Teacher Forcing scheme, including decoder-only Autoregressive LM finetuned on GPT-2 (**FT**), encoder-decoder (**Enc-Dec**) transformer model, and Varitional Information Bottleneck (**VIB**) (Alemi et al., 2016). We initialized the FT model using GPT-2 large (774M), whereas encoder and decoder in the Enc-Dec/VIB models (695M/697M) are initialized with bert-large and GPT-medium, respectively. All the considered models are finetuned on the target datasets. We follow Li et al. (2022) to report the FT baselines with two decoding strategies, top-p sampling (K=50, p=0.92) and beam search (beam width 4), denoted as FT-sample and FT-search. We use top-p sampling for Enc-Dec/VIB generation. For summarization tasks, we finetune a T5-large model (770M) on the target datasets as baselines. We also compared two text diffusion models Diff-LM and Genie using their suggested configuration from the official repository. More details are in App. F.

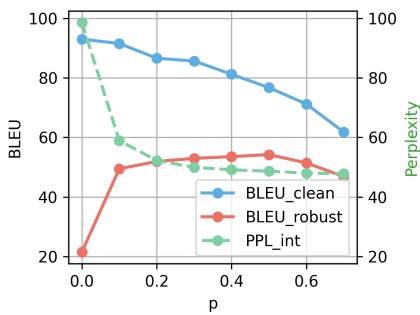

Figure 3: Impact of the proportion of injected noise for learning Paragraph Embeddings on XSum dataset. Large substitution noise results in worse **BLEU$_{\text{clean}}$** but better **BLEU$_{\text{robust}}$** and **PPL$_{\text{int}}$**.

# 6 Results

## 6.1 Paragraph Representation Learning

It is essential to learn a paragraph embedding space that is both *accurate* and *smooth*. To this end, we examined the effect of various substitution probabilities $p$ to the input tokens $\boldsymbol{x}$. Our findings, presented in Fig. 3, reveal that a smaller $p$ results in a lower conversion error, as indicated by a higher reconstruction BLEU (**BLEU**$_\texttt{clean}$), albeit at the expense of local smoothness (**BLEU**$_\texttt{robust}$) and distributional smoothness (**PPL**$_\texttt{int}$). We performed a grid search of $p$ with $0.1$ increment based on $512$ samples. Empirically, we observed that a weighted score $S_\text{overall} = 0.5\textbf{BLEU}_\texttt{clean} + 0.8\textbf{BLEU}_\texttt{robust} - 0.3\textbf{PPL}_\texttt{int}$ correlate well with downstream diffusion performance, leading to fluent and accurate generation for PLANNER (see App. A for more details). We finally opted for $p = 0.3$ for most datasets [2], which strike a balance between conversion error and smoothness.

It is worth noting that there is an inevitable conversion loss, indicated by the fact that the **BLEU**$_\texttt{clean}$ is between $77 \sim 87$ when generating hundreds of words (App. D). We observd that most of the lexical mismatch still maintain the similar semantics, with the exception of some name entity swapping. We show some paragraph reconstruction and denoising examples from our paragraph embedder in the App. A. We also include examples of interpolated generation from random paragraph pairs in the App. A. In general the transition of semantics is natural, indicating reasonable distributional smoothness of the paragraph embedding space.

## 6.2 Sentiment-Guided Generation

For sentiment-guided generation experiments, following previous works (Li et al., 2022; Hu et al., 2017; Keskar et al., 2019), we use a trained classifier to assess if the given sentiment is well-controlled in the generation. The trained classifier is initialized with BERT-large and finetuned on the training set, which yields an accuracy of $96.75\%$ on the held-out test set.

| Arch. | PPL | ACC↑ | DIST/ENT↑ | S-BL↓ | Rep-4↓ | Len |
|-------|-----|------|-----------|-------|--------|-----|
| FT-sample | 20.86 | 70.2% | 0.13/6.154 | 0.96 | 5.86% | 113 |
| Diff-LM | 101.97 | 83.6% | 0.15/5.115 | 4.05 | 6.23% | 66.2 |
| Ours | 51.12 | **94.9%** | **0.16/6.360** | **0.77** | **2.37%** | 161 |
| Human | 47.94 | 96.7% | 0.17/6.737 | 0.48 | 2.17% | 157 |

Table 1: PLANNER achieves high success rate (**ACC**) and diversity with less repetion in generating hotel reviews conditioned on sentiment.

The results are provided in in Tab. 1. PLANNER outperforms the baseline approaches in generating long reviews at higher levels of accuracy. Although PLANNER using a greedy decoding mode is at least comparable with FT with top-p sampling in terms of diversity, and has lower repetition as assessed by Rep-4 in generation.

The perplexity of the text generated by PLANNER is close to that of human-written text. We provide examples of the generated text in App. E. Interestingly, as shown in App. E, with the same random seed but different controlling sentiment, PLANNER generates text with similar contents but different sentiments, suggesting the diffusion model may be able to disentangle the semantic space to certain extent. Unlike the autoregressive generation, the nature of the diffusion model allows the model to "regret" and iteratively refine on the current generations. In App. B, we demonstrate how the generation evolves over multiple time steps in a coarse-to-fine manner in PLANNER.

## 6.3 Long-form Text Completion

We further evaluate our model on the long-form text completion tasks. For text diffusion baseline, we compared our method with Diff-LM (Li et al., 2022) on hotel review dataset. We could not perform a comparison on the C4 dataset with Diff-LM due to the significant amount (thousands) of GPU hours required to train Diff-LM adequately. A Diff-LM running time estimation is available in App. F. The results are provided in Tab. 2. FT-search performed poorly in this open-ended generation task as its generation exhibited high repetition levels, consistent with findings in previous research (Holtzman et al., 2019; Xu et al., 2022). Although our approach also employs a deterministic decoding method, we observed that it produces text with low Rep-4 metric, signifying that PLANNER is effective in

---

[2]except for CNNDM dataset where we use $p = 0.5$

| Arch. | PPL | DIST/ENT↑ | S-BL↓ | Rep-4↓ | BL↑ | R-L↑ | Score↑ | Len | AuBL↑ |
|---|---|---|---|---|---|---|---|---|---|
| | | | | *Hotel Review* dataset | | | | | |
| FT-search | 1.87 | 0.03/4.865 | 3.50 | 86.60% | 0.62 | 5.2 | 0.39 | 179.51 | - |
| FT-sample | 15.51 | 0.14/6.455 | 0.88 | 4.49% | 0.78 | 6.8 | 0.53 | 164.50 | - |
| Enc-Dec | 33.82 | 0.18/6.379 | 0.57 | 3.25% | 0.47 | 7.3 | 0.54 | 94.03 | |
| VIB | 36.89 | 0.19/6.481 | 0.54 | 3.15% | 0.45 | 7.1 | 0.54 | 86.11 | - |
| Diff-LM | 178.30 | 0.13/5.560 | 3.57 | 4.54% | **0.84** | **8.8** | 0.43 | 175.10 | 26.16 |
| PLANNER | 47.36 | **0.17/6.602** | **0.52** | **1.55%** | 0.77 | 7.9 | **0.55** | 168.08 | **38.55** |
| Human | 47.60 | 0.20/7.023 | 0.60 | 1.46% | - | - | - | 181.29 | - |
| | | | | *C4 subset* dataset | | | | | |
| FT-search | 1.927 | 0.07/6.245 | 0.14 | 79.54% | 0.77 | 5.2 | 0.37 | 154.88 | - |
| FT-sample | 12.244 | 0.25/7.136 | 0.44 | 7.01% | 1.59 | 5.9 | 0.47 | 122.55 | - |
| Enc-Dec | 23.095 | 0.24/7.077 | 0.16 | 2.27% | 1.92 | 7.5 | 0.5 | 118.07 | - |
| VIB | 19.701 | 0.24/7.003 | 0.16 | 2.62% | 1.86 | 6.8 | 0.49 | 113.34 | - |
| PLANNER | 61.768 | **0.28/7.352** | **0.12** | **1.67%** | **2.04** | 7.7 | **0.51** | 111.89 | 36.77 |
| Human | 59.783 | 0.44/7.381 | 0.12 | 1.12% | - | - | - | 107.56 | - |

Table 2: PLANNER enhances the diversity of text generation and minimizes the occurrence of repetition in open-ended text completion tasks.

reducing repetition through holistic iterative refinement throughout the inference steps in the diffusion process. Comparing with Diff-LM and other baselines, PLANNER achieved better diversity scores while maintaining comparable relevance scores. We also observe higher AuBLEU of PLANNER comparing with Diff-LM, indicating a potentially higher overall denoising strength of PLANNER (See App. D for more details). Some examples of the generated text are available in the App. E. We also observed PLANNER exhibits robustness towards prompts that are either repetitive or ill-composed, where FT failed (Fig. 1, App. G).

We further performed pairwise human evaluation on 300 examples of hotel review generation from each system on our internal crowd-source annotation platform. Each pair of text being presented to 3 judges in random order. The judges ranked the pairs for relevance, informativeness and human-like properties using a 3-point Likert-like scale. Overall judge preferences are shown in Table 3. A moderate preference can be observed for PLANNER over FT and VIB, except for human-like between PLANNER and VIB. We also observe that judges still prefer human responses over system generations in this task. Further details, including the human evaluation template used and interrater agreement analysis, are provided in the App. H.

| Metric | Methods | Win | Tie | Loss |
|---|---|---|---|---|
| Rel. | Ours vs. FT | **48.2**% | 9.2% | 42.6% * |
| | Ours vs. VIB | **50.7**% | 10.0% | 39.3% ** |
| | Ours vs. Human | 39.3% | 9.3% | **51.3**% ** |
| Inf. | Ours vs. FT | **55.1**% | 5.7% | 39.2% ** |
| | Ours vs. VIB | **48.7**% | 8.0% | 43.3% * |
| | Ours vs. Human | 37.7% | 8.7% | **53.7**% ** |
| Hum. | Ours vs. FT | **51.5**% | 8.4% | 40.1% ** |
| | Ours vs. VIB | 40.0% | 19.3% | **40.7**% |
| | Ours vs. Human | 34.3% | 17.0% | **48.7**% ** |

Table 3: Human evaluation on Relevance (Rel.), Informativeness (Inf.), and Human-likeness (Hum.). Statistical significant results: ** $p < 0.001$, * $p < 0.01$.

## 6.4 Summarization

We further conducted evaluation on summarization and present the results in Tab. 4. Summarization is less open-ended than the text completion task, thus a deterministic decoding approach like T5-search can produce high-quality text. Our evaluation shows that in comparison with T5-sample and Genie (Lin et al., 2022), PLANNER exhibits comparable Rouge-L scores, while improves other metrics. PLANNER achieves higher AuBLEU than Genie (See App. D for more details).

Owing to the sampling nature of the diffusion model, PLANNER and Genie yielded lower Rouge-L scores in comparison with T5-search, with single summary. To align with Genie's evaluation, we provide the results with 10 random runs in Tab. 4, where for each document it generates 10 summaries and the one with the highest Rouge score is used. However, we note that these summaries with

best Rouge-L cannot be predetermined without an oracle summary. Comparing with T5-search, PLANNER generates more diverse and less repetitive summaries. However, the improvement is less conspicuous comparing with the results observed in open-ended text completion tasks. We show some generations in the App. E (Tab. 12).

Notably, the generated content may occasionally include hallucinations or errors, especially for name entities and digits (App. E, Tab. 13). Such occurrences can be attributed to either the conversion errors in $\mathcal{D}$ or errors during the generation of paragraph embeddings, and requires further investigation.

| Arch. | PPL | DIST/ENT↑ | S-BL↓ | Rep-4↓ | BL↑ | R-L↑ | Score↑ | Len | AuBL↑ |
|---|---|---|---|---|---|---|---|---|---|
| | | | | *CNN Dailymail* dataset | | | | | |
| T5-search | 58.12 | 0.11/7.726 | 0.24 | 6.69% | 7.66 | 34.48 | 0.66 | 45.51 | - |
| T5-sample | 67.58 | 0.11/7.790 | 0.20 | **3.50%** | 5.05 | 30.15 | 0.64 | 48.51 | - |
| Genie | 179.9 | 0.09/7.293 | 0.24 | 4.16% | 3.22 | 30.47 | 0.58 | 40.94 | 27.21 |
| Genie[10] | 170.6 | 0.10/7.355 | 0.24 | 4.32% | 6.48 | **37.09** | 0.62 | 40.81 | - |
| PLANNER | 49.21 | **0.10/8.037** | **0.15** | 5.25% | 6.92 | 30.43 | 0.62 | 52.33 | **43.91** |
| PLANNER[10] | 49.07 | 0.10/8.019 | 0.15 | 4.96% | **11.42** | 36.81 | **0.66** | 53.14 | - |
| Human | 49.477 | 0.12/8.226 | 0.16 | 5.63% | - | - | - | 51.15 | - |
| | | | | *XSum* dataset | | | | | |
| T5-search | 29.41 | 0.12/7.200 | 0.31 | 14.83% | 6.11 | 36.08 | 0.74 | 18.97 | - |
| T5-sample | 36.17 | 0.13/7.449 | 0.24 | 6.47% | 3.62 | 31.18 | 0.71 | 20.78 | - |
| Genie | 186.7 | 0.09/6.935 | 0.28 | 8.56% | 2.38 | 34.85 | 0.66 | 20.44 | 30.85 |
| Genie[10] | 178.2 | 0.09/6.924 | 0.30 | 9.66% | 5.06 | **41.59** | 0.68 | 19.97 | - |
| PLANNER | 67.94 | **0.11/7.553** | **0.21** | **5.38%** | 4.84 | 33.97 | 0.69 | 20.04 | **57.88** |
| PLANNER[10] | 67.46 | 0.11/7.529 | 0.23 | 5.82% | **11.61** | 41.23 | **0.72** | 19.89 | - |
| Human | 37.8 | 0.13/7.656 | 0.21 | 5.56% | - | - | - | 21.19 | - |

Table 4: For summarization task, PLANNER outperform Genie (Lin et al., 2022) in generation diversity and fluency while maintaining comparable Rouge-L scores. [10] indicates the maximum results after 10 runs, following (Lin et al., 2022).

## 6.5 Analysis

**Running time** We conducted inference time benchmarks of each method on a single Nvidia A100. For the sentiment-guided generation task, the autoregressive baseline is 5x faster than our method as the generation for all methods can be batched. For all other tasks, the varying input lengths make direct batchification for the FT baseline not straightforward. In these scenarios, the latent diffusion over a fixed number of latent codes offers computational advantages over a naive decoding of the FT baseline as the latent codes in our method can be conveniently batched.

For the hotel review completion task, the generation of 256 samples took 378 seconds to complete, including 83 seconds for decoding and 295 seconds for diffusion generation with 30 generation steps. The unbatched FT baseline took 1,693 seconds to complete 256 generations. Sorting input text by length and maximally batchifying them as possibley reduce the (batched) FT inference time to 338 seconds. The Diff-LM algorithm required 397 seconds to produce 256 samples using 200 generation steps, which is comparable to ours. Although our method is slower than the autoregressive ones, PLANNER enjoys the convenience of arranging input into the same length vectors without further length bucketing. On the CNN-DM summarization tasks, our method took 8.4 GPU hours to generate 11392 summaries. Genie's generation took 47.2 GPU hours. XSum gives similar inference running time benchmark to the results on CNN-DM.

**Generations over diffusion steps** In App. B we provide generation examples for both summarization and sentiment-guided generation over different diffusion steps, which progress in a coarse-to-fine manner. The generation from early time step tends to be less fluent and generic. As the time approaches 0, the generation becomes more detailed. We presented quantitative results characterizing the evolution of the metrics over generation steps in App. B, Fig. 5. It revealed a clear trend of improvement in the majority of the metrics as the generation proceeds. Notably, most hallucinations

occur during the late phase when more details are being incorporated. The model may excessively emphasize certain aspects, resulting in the correct generation being altered to an erroneous one (App. B, Tab. 10).

## 7 Conclusion

We present a two-stage latent text diffusion model that uses an autoencoder to condense lengthy texts into a limited number of paragraph embeddings, and a continous time diffusion model that learns the distribution of these embeddings. Our proposed model alleviates the issue of repetition and advances generation diversity across different tasks. Compared to text diffusion models that perform diffusion solely on token or token embedding space, our method generates fluent text with improved diversity and reduced repetition. There may be toxicity or fairness issues in the dateset we used that we have not been able to identify. There are several limitations that warrant further investigation. Our work relies on an autoregressive decoder for converting latent representation into coherent text. It is possible to explore the feasibility of non-autoregressive decoders to bolster efficiency while minimizing conversion errors and hallucination in the generation. Furthermore, the classifier-free guidance approach results in a discrepancy between training and inference data distribution when feeding to the diffusion backbone. It would be interesting to investigate a "calibration" strategy for the latent code to better fit the data distribution during training.

## Acknowledgement

We thank Yinfei Yang, Barry Theobald, Zhe Gan, Edouard Grave, David Grangier, Tatiana Likhoma-nenko, Richard Bai and Ronan Collobert for their critical suggestions and helpful feedback throughout this project.

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
