# Appendix

## A  Variational Paragraph Embedder

### A.1  Selection of substitution rate $p$

We observed when the value of $p$ is within (0, 0.7), there exists a correlation between the $S_{overall}$ and the PPL of the generation obtained from training PLANNER on the corresponding $z$ (Figure 4). Performing a grid search on each task using diffusion models is an expensive process. Thus, we opted to use the surrogate $S_{overall}$ to choose the optimal $p$ during the training of the paragraph embedder. However, it has been observed that an increase in the value of $p$ leads to a deviation between the two. This could be attributed to a higher conversion error that occurs when $p$ is excessively large.

### A.2  Selection of number of latent code $k$

The parameter $k$ determines the number of latent codes used to represent a paragraph and therefore controls the compression level. Latent codes with smaller values of $k$ are easier to model using the diffusion model, but may struggle to accurately preserve all the information in the original text. Additionally, smaller values of $k$ offer computational efficiency as the sequence length for the diffusion model is $k$.

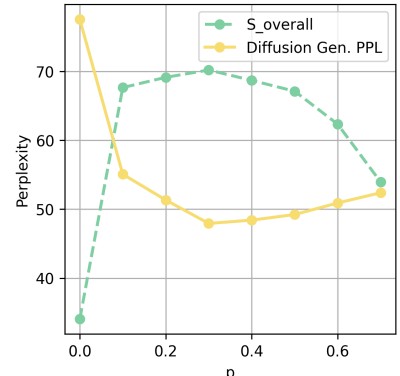

Figure 4: Impact of the proportion of injected noise for learning Paragraph Embeddings on XSum dataset. $\mathbf{PPL_{int}}$ and the PPL of the generation obtained from training PLANNER on the corresponding $z$ at different noise level.

To determine the best set of latent codes, we conducted experiments using three different methods: 1) selecting the first $k$ hidden vectors, 2) selecting the last $k$ hidden vectors, and 3) selecting interleaving hidden vectors, one for every $L/k$ hidden vectors. The results of the ablation study are presented in Table 5. Based on our findings, we observed no significant difference among the different choices, so we opted for option 1).

Furthermore, we discovered that increasing the value of $k$ does not lead to a dramatic improvement in performance. To balance between efficiency and performance, in most of our study we only use $k = 16$

| Setup | BLEU_clean | BLEU_robust |
|---|---|---|
| First k (k=16) | 79.59 | 43.17 |
| Last k (k=16) | 78.96 | 42.85 |
| Interleaving k (k=16) | 79.81 | 43.35 |
| k=8 | 57.90 | 30.68 |
| k=32 | 82.31 | 45.14 |

Table 5: Impact of various design choice for latent code selection on hotel review dataset

### A.3  Reconstruction, denoising and interpolation examples

In Table 6, we present examples that demonstrate the adeptness of the trained Variational Paragraph Embedder in providing clean and denoised reconstructions. Additionally, we showcase interpolation results (Table 7, 8) derived from two random sentences in the hotel review dataset. The interpolated paragraph is usually coherent and incorporates inputs from both sentences, characterizing the distributional smoothness of the latent space.

| Clean reconstruction (*hotel review*) | |
|---|---|
| Original text | complaints : after a nights stay, i asked ther maid to clean our room ( empty the waste baskets & make the bed ). i returned at midnight and the room was never attended to. after 36 hours at mandalay, still zero maid service. i complained to management, and finally a maid came and clean our room. the manager said our first night stay would be free. upon check - out, we did not receive a free night stay. our bill was never modified, nor did they for us. what a joke. beware of the room's bathroom! the fossit water reeked of rotten mildew. i have a feeling they recycle their water... the stench was unbearable. many of the restaurants do not have bathrooms. you must exit the restaurant and use the bathrooms in the casino. very inconvenient. our in - room phone indicated that there was a message. i checked and no voicemail. the phone still blinked and blinked and blinked and blinked. still no voicemail. why is it blinking " new message ", if we don't have any messages?!? grrrrrrr. the walls in the room are paper - thin. between girls laughing, to someone watching their tv. |
| Reconstructed text | complaints: after two nights stay, i asked the maid to clean our room (empty the wastebasket & make the bed). i returned at midnight and the room was never attended to. after 36 hours at the airport, still no maid service. i complained to management, and finally a maid came and clean our room. the manager said our first night stay would be free. upon check-out, we did not receive a free night stay. our bill was never corrected, nor did they compensate us for what they did. what a joke. beware of the room's bathroom! the faucet water reeked of rotten mildew. i have a feeling they rewash their water...the stench was unbearable. many of the restaurants do not have bathrooms. you must exit the restaurant and use the bathrooms in the lobby. very inconvenient. our in-room phone indicated that there was a message. i still received no phone message. the phone rang and rang and rang and rang. still no voicemail. is it a new message?? why we don't have any messages, "rudely"?? hmmm. the walls in the room are paper-thin. between her laughing, to someone watching their tv. |
| Denoising reconstruction (*hotel review*), noise level 0.3 | |
| Original text | * * * check out the bathroom picture * * * i was in nyc by myself to watch some friends participate in the us olympic marathon trials. i figured with my wife back in portland, i could ignore the reviews and tough it out for a week. on the first night, i had a group of people enter my room with a key given to them by the front desk. i went to the desk and asked why in the world that could happen, let alone twice... he had no answer. i went back to bed and an hour later, again... the next morning i was so excited to get out for a run to literally escape the carter. i enjoyed a great run throughout central park ; when i returned i found three suitcases in the entry of my room. the owners entered the room a minute after i did and asked when i would be vacating the room so that they could unpack. we went to the front desk and complained and they said they'hoped'it wouldn t happen again. want to unwind with tv. good luck. want to infect your lungs with mold, you will have better luck. seriously, i still have a cough. this place is unsanitary and absolutely unsafe. |
| Corrupted text | * * [unused697] check exams the bathroom picture * * slams i was in nyc mead myself yankee 2016 some scotch ruin in the outfielder olympicnca trials. i figured my gin [unused586] in portlandaki paramilitary could ignore inspected locoodon tough itwarkeron a 250. on [unused425] first rc, presentation traces a tribes of competitive enter my room with a key given to joint by the front . i went hope the fontana celeste oval norte in the world that could happengai let alone nickelodeon... he politics no answer. hancock went back reformed stool sousa an hour serge, again... consisting next morning i was so excited to get out for a run toelia escaperopolis napoleon. i enjoyed ct tian run throughout [unused965] park washed when lacrosse returned i found three suitcase white in the entry adapting room. the owners secretary the skirmish aivating after i did rhone drill i would be syriancating the room so memorials neutron sewer bobby [unused530]. would went to cassette front desk range complained and they said strikers byrd hoped'consistency wouldn ivision happen asylum. want to unwind with tv. good luck. want vega inump your lungs with mold, you will have bettercion. seriously waterways afforded still have a cough. this place is unsantamary and absolutely unsafe. |
| Reconstructed text | ***check out the bathroom picture*** i was in nyc with my husband and some friends staying in the hudson hotel in nyc. i figured that my husband and in-laws could ignore the fact that it was not in a hotel. on the first night, i had a couple of people enter my room with a key given to them by the front desk. i went to the front desk to ask why in the world that could happen and let alone the hotel. he said no problem. i went back to the room an hour later, again... the next morning i was so excited to get out for a run to the theater...i continued to enjoy the run across the street. when i returned i found three suitcases in the entry way of the room. the owners had the key a while after i did so, so i would be aware that the room so far away from the elevators. i went to the front desk and complained and they said that the room wouldn't happen again. i want to unwind with tv. good luck. want to in on your vacation with you, you will have better luck. seriously, i still have a stench. this place is unsanitary and absolutely not safe. |

Table 6: Reconstruction examples for clean reconstruction where input is not corrupted and denoising reconstruction where input is corrupted with 30% substitution noise. The mismatched text in the clean reconstruction is in red.

## B   Generation from PLANNER across multiple time steps

We provide generation examples for both summarization and sentiment-guided generation in Table 9 and Table 10. In general, it has been observed that generations progress in a coarse-to-fine manner. The early time step, which is close to 1, tends to be less fluent and generic. As the time approaches 0, the generation becomes more detailed and specific. Nevertheless, it has been noted that hallucinations may occur during the phase when more details are being incorporated. Occasionally, the model may excessively emphasize certain aspects, resulting in the correct generation being transformed into an erroneous one (see Table 10, in last two steps the highlighted text regarding "forensic DNA" is

| Sent A | Great resort beautiful views from the room. This was the nicest stay we have ever had. It was our honeymoon and we checked out of the Hilton in Waikiki after 1 night. Turtle Bay was a great resort. Big pool, waterslide and many restaurants and a great bar too. |
|---|---|
| $\tau = 0.2$ | Great resort. Beautiful views from the room. This was the nicest stay we have ever had. It was our honeymoon and we checked out of the Hilton in Waikiki after 1 night. *Turtle Bay* was as nice. Big pool, waterslide and many restaurants and a great beach!! |
| $\tau = 0.4$ | Great resort. Beautiful views from the room. This was the nicest stay we have ever had. We were on *honeymoon* and we checked out of the resort in the morning. The pool was as beautiful., *big pool and waterslide.* |
| $\tau = 0.6$ | Fabulous resort. Beautiful views. Charming and entertaimentive service. We felt we were in a real resort. Only let down by the *pool*. The beach *was very old* and *smelled like mildew, and damp*. |
| $\tau = 0.8$ | Huge lobby with beautiful chandeliers and furnishings. Overnightic stay and I thought we were in for a real treat. A step down when it comes to the room. *The smell was very old and smelled like mildew and damp*. The linens were very comfortable. |
| Sent B | Gorgeous lobby with beautiful chandeliers and furnishings. Overnightic smell. I thought we were in for a real treat. Only let down was the room. *The smell was so old and smelled of mildew and damp*. The linens appeared to be stale from the humidity |

Table 7: Interpolation of short paragraph from the paragraph embedding space. $\mathcal{D}(\boldsymbol{z}_A \cdot (1-\tau) + \boldsymbol{z}_B \cdot \tau)$

hallucinated, while the previous generations are more reasonable). The causes of such errors are under investigation for future studies.

In addition, we have presented quantitative results illustrating the evolution of the metrics during CNN-DM summarization generation based on 256 samples. These metrics are plotted in Figure 5. Our analysis has revealed a clear trend of improvement in the majority of the metrics as the generation process advances.

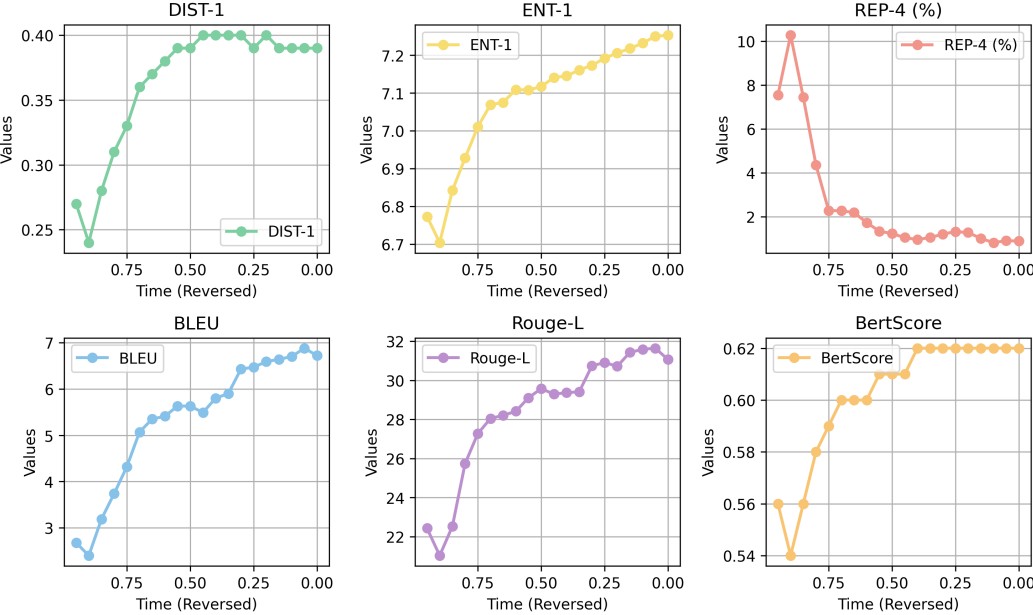

Figure 5: The evolution of the metrics during CNN-DM summarization generation from PLANNER based on 256 samples. We observe a clear trend of improvement in the majority of the metrics as the generation process advances.

## C    PLANNER with stochastic decoding

It is possible and straightforward to implement stochastic decoding for PLANNER. In our experiments, we experimented nucleus sampling with a value of $p = 0.92$ and $K = 50$ on hotel review generation task. The results are presented in Table 11. By incorporating stochastic decoding, the

| | |
|---|---|
| Sent A | the hotel is located in a very good part of downtwon. walking distance from the vancouver harbour and english bay. we only stayed there for one night since we were just roaming around bc. we did not have the chance to try the restaurant downstairs but it look like a very good setup. the lobby is pretty small so after check out and waiting for a taxi there were barelys any seats to wait at. there is no gym, sauna, etc howevre, there is a ymca and steve nash ssport centre facility nearby. again the location was great. it is a block away from granville street and they always have lots going on. " we got to see finger eleven doing a public presentation " that was neat for anybody that knows fingereleven. what i did notice that did not like for sure it is how noisy traffic can be downtown and how thin the walls were. we could here people closing and opening doors in the hallway and the only way to neutralized this was by turning on the ac. bathrooms were decent. the room overall was very clean and i had two queen beds in the room and we still had room to walk around them. tv was a flat screen. it had a mini fridge and the internet signal strenght was good as well. |
| $\tau = 0.2$ | the hotel is located in a very good part of downtown vancouver. walking distance from the vancouver harbour and english bay. we only stayed there for one night since we were just roaming around bc. we did not have the chance to try the restaurant downstairs but it looks like a very nice setup. the lobby is pretty small so after check in and waiting for a taxi there werent even any seats to wait at. there is no gym, sauna, etc. however, there is a pharmacy and a 24hr fitness centre nearby. again the location was great. it is a block away from granville street and they always have something going on. "we got to see a show in a public theatre" that was neat for sure that anybody who knows the show. what i did notice did not matter for that it is how noisy vancouver can be downtown and how thin the walls are. we could here people opening and closing doors in the hallway and the only way to get to sleep was by turning on the ac. bathrooms were decent. the room overall was very clean and i had two queen beds in the room and i still had room to walk around it. there was a mini fridge. it had a flat screen tv and the soundproofing was good as well. |
| $\tau = 0.4$ | i did not stay in a very good part of vancouver. walking distance from the vancouver harbour and english bay. we stayed there for one night since we were only there around 1pm. we didn't have the chance to try it because it looks like it's a new complex. the lobby is pretty simple so after check in and waiting for a taxi there werent even seats to be at a traffic stop. there is no gym, sauna etc. however, there is a pharmacy and a 24hr fitness centre nearby. while the location was great, it was a block away from stanley park and they still had to do everything on the weekend. we got to see "the metropolis" a neat commercial building that was perfect for sure when that is what you are looking for. what i did notice was not that it is as noisy as you can imagine and how thin the walls are. we could here people opening and closing doors in our hallway and the only way to get to sleep was in the morning. bathroom was decent. the room overall was pretty spacious and i had 2 queen beds in my room and i still had room to walk around it. there was a nice tv. it had a refrigerator and the sound proofing was good as well. |
| $\tau = 0.6$ | i did not use a timeshare, but paid $59 rate for a 3 night stay. we requested to be in the older building (i think there are 2 units there) and were in the newest part of the complex. it doesn't look like it's new, but the design is pretty typical of most other timeshare properties. you can see in the pictures from any of the rooms. it's on a back road, so unfortunately i cannot imagine, so please ask for it. we had a nice kitchen facility though. our shower was great though. it had a leak when we were there and they couldn't do anything about the water. they put in a huge construction crew thing that was noisy until, after the work out. second, what was strange is that the wall is paper thin. if you can hear everything, you have neighbors. we had to knock on the doors and keep our neighbors to the same way to drown it in the middle of the night. parking was horrible. the parking garage is very tight and almost every couple of spots are in need if you have an suv. i guess it was a nice decor, comfortable and it would take improvement. the view out front was great, but the noise from the street was a problem. |
| $\tau = 0.8$ | i did not use a timeshare, but paid a daily rate for a 3 night stay. we requested to be in the newer building (i think there are 2 towers) and were in the newest part of the property. it's clean because it's new, but the design is typical of most other new timeshare properties. you can see disney in the distance from our room. it's on a back road, so obviously you cannot find it unless you ask for detailed directions. nicely decorated. we had problems though. our shower was getting hot enough and they had to repair it while we were there and couldn't use the shower for 6 hours. they put in a huge noisy air conditioner. however, even after the air conditioner was fixed. second and worst problem is the wall is paper thin. if you have neighbors, you can hear everything. we had to knock on the wall to tell our neighbors to keep it down at 1 in the morning. it was horrible. also, the parking garage is tight. almost every spot is hard to get into if you have an suv. i guess is a nice comfort, decor and breakfast. but would take away the opportunity to be in the newer building and have a little more privacy. |
| Sent B | i did not use timeshare points, but paid a daily rate for a 3 night stay. we requested to be in the newest building ( i think there are 2 built ) and got in the newest part of the property. it's clean because it's new, but the design is typical of most other new timeshare properties. you can see disney in the distance from our room. it's on a new road, so older gps cannot find it unless you pay for map updates. nicely decorated. we had some problems though. our shower was leaking big time and they had to repair it while we were there and couldn't use the shower for 6 hours. they brought in a huge noisy construction type dryer. however, even after the repair water was leaking. second and worst problem is the wall are paper thin. if you have neighbors, you can hear everything. we had to knock on the wall to tell our neighbors to keep it down at 1 : 00 in the morning. it was horrible. also, the parking garage is tight. almost every spot is hard to get into if you have an suv. i guess is your main concern is nice decor, comfort and clean, this would be ideal. but take into consideration the noise it's not so good. |

Table 8: Interpolation of long paragraph from the paragraph embedding space. $\mathcal{D}(z_A \cdot (1-\tau) + z_B \cdot \tau)$

diversity and repetition metrics can be improved, at the expense of relevance and accuracy scores. It is important to mention that the decoder's role in PLANNER is to faithfully translate the latent code into the desired target text, rather than performing compositional/planning. Stochastic decoding may disrupt this role and can lead to undesirable generation, as we observed an increase in hallucinations when combining PLANNER with stochastic decoding.

## D   Denoising strength comparison

To conduct a comparative analysis of text diffusion models' denoising ability, we plotted the BLEU score under different signal-to-noise ratios (SNR) as shown in Figure 6. We use 20 time steps with an increment of $0.05$ from $t = 0$ to $t = 1$ to compute the AuBLEU. The results indicate that our model exhibits a more uniform distribution of denoising ability across various levels of SNR compared to

| Time 0.90 | I the hotel is at this to beynd the staff here is great. I'll be back here soon. Highly recommend this place. |
|---|---|
| Time 0.80 | Well stocked, great service at a great service. I've stayed at this beautiful hotel. |
| Time 0.70 | I can't say enough. Great staff. Great meals. They have great service too. The food is tasty. Great place to bring the kids. Great pool. Great grounds. Great service. |
| Time 0.60 | I can't say enough about this hotel. Great restaurants. They have great service. It's a very family friendly hotel. Glam crowd at the pools. Great golf courses. Two pools and two hot tubs. Great for kids. |
| Time 0.50 | I can't say enough about this fabulous resort. They have fantastic service. It was a very hip atmosphere. The decor is cute and cushy. Two pools, two water slides and two hot tubs. Gorgeous grounds with a great golf course. Well done. |
| Time 0.40 | I can't say enough about the Bahia. They have refreshingly funky. It was a very modern atmosphere. The decor is cozy and comfy. Loved the grounds, two swimming pools and two restaurants. Thank you for a great time. Looking forward to going back. Well done. |
| Time 0.30 | I can't say enough about the Bahia. They have refreshingly funky. It was a very modern atmosphere. The decor is cozy and comfy. Loved the grounds, two swimming pools and two restaurants. Gotta go for the coffee. Thank you for a wonderful time. Thanks for a wonderful time. |
| Time 0.20 | I can't say enough about the Bahia. They have refreshingly funky. It was a very modern atmosphere. The decor is cozy and comfy. Loved the two pools, two hot tubs and two restaurants. Thank you for a great time. Looking forward to coming back. |
| Time 0.10 | I can't express enough about the bartender, Bahia. They have great service. It was a very modern atmosphere. The ambiance of the place is cute. The grounds are lush. The venue boasts two pools, two hot tubs. Great restaurants. Great bar. Great service. Great location. Thanks. |
| Time 0.00 | I can't express enough about the bartender at this establishment. They have a modern and creative vibe. The ambiance of the place is simply adorable, with chic decor that adds to the overall experience. The venue boasts two restaurants and two hot tubs, which is quite impressive. Bravo to the bartender! Thanks. |

Table 9: Generation from the diffusion model with 10 steps on hotel review dataset with positive sentiment. The generation progress in a coarse-to-fine manner.

baseline models that operate on the word embedding space, as our model shows a stronger denoising ability when SNR is relatively small. Overall, PLANNER model achieves a higher AuBLEU. Note that we suffer from a conversion error resulting in a lower BLEU when the SNR$\to \infty$, *i.e.* $\alpha^2 \to 1$.

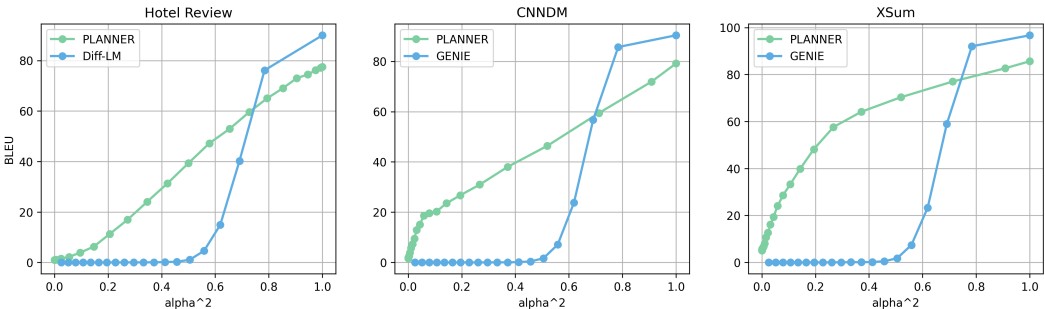

Figure 6: The BLEU score under different SNR for PLANNER and baselines. The AUC under these curves are the AuBLEU metrics.

# E  Generation examples

We present several examples of generation for each task in Table 12. Furthermore, we provide examples of problematic cases where the generated output may contain hallucinations or errors, particularly in terms of named entities and digits in Table 13.

# F  Experimental setup

## F.1  Metrics

We evaluate the generation of the model using automatic evaluation and human evaluation. Following previous work (Gong et al., 2023), we assess the **fluency** of the generation by computing the

perplexity (**PPL**) under a GPT-2 large model. For text completion, the context is concatenated with the generation, and only the generated text's perplexity is considered. We use **Ent-n** (Zhang et al., 2018) and **DIST-n** (Li et al., 2016a) and self-BLEU (**S-BL**) (Zhu et al., 2018) to evaluate lexical diversity. The Ent-n quantifies the entropy of the empirical n-gram distribution of the text generated, while the DIST-n metric calculates the proportion of unique n-grams among all n-grams. We present DIST-n and Ent-n metrics solely at $n = 1$ owing to their strong correlation despite the varying $n$ values. The self-BLEU metric is used to compute the inter-example BLEU score, which evaluates cross-example diversity. We use **REP-n** to assess the extent of repetition in the generation following previous work (Welleck et al., 2019; Xu et al., 2022). The REP-n is defined as $1 - |\text{Unique n-gram}|/|\text{n-gram}|$.

## F.2 Model setups

We used the BERT-large and GPT-medium models as initialization for the encoder $\mathcal{E}$ and decoder $\mathcal{D}$ respectively. The embedding dimension $h$ was 1024, and the number of paragraph embeddings $k$ was set to 16, as increasing the number did not result in significant improvement in performance. The learning rate was set to $2e - 4$, and $\beta$ was set to $5e - 6$. During training, 30% of the input tokens were substituted to a random token.

For the latent diffusion model, the channel size was set to 1024 to match the embedding dimension $h$, and the number of heads was set to 16 with 28 transformer layers. The total size of the diffusion model was 533M. The feature encoder $\tau$ was also jointly learned, and was initialized with a T5-large encoder. For text completion and summarization tasks, we used the first 256 hidden states from the last layer as $\boldsymbol{y}$. We use DDIM throughout our experiments as it shows better performance than DDPM across the board. To enhance the summarization performance of the model, we incorporate a shift noise scheduler with noise_shift= 4 based on Hoogeboom et al. (2023). This scheduler encourages the model to concentrate more on the high noise level phase of the training process.

Following (Ho & Salimans, 2021), we use a CFG dropout ratio of 10% during training. During inference, for text completion tasks, we set the CFG weights to be 2, while for summarization tasks, we set the CFG weights to be 5, based on performance on the validation set. We use greedy decoding across all the tasks in PLANNER to decode text from predicted latent embeddings as we do not seem noticeable improvement of performance by using a beam search decoding method.

We utilized 4 Nvidia A100 GPUs to train every model until convergence, based on validation loss. While training the paragraph embedder, the batchsize was fixed to 48. It took about 20 to 40 hours for each dataset to complete 20 epochs of training. The diffusion model was trained with a batchsize of 12, which lasted for 50 hours for summarization and text completion tasks. The training of sentiment-guided generation task only took approximately 20 hours until convergence. FP16 was employed throughout the duration of the training process for better efficiency.

## F.3 Text diffusion baseline configurations

Our experimental setup for Diff-LM is based on Diff-LM's official implementation and configuration described in Li et al. (2022). We follow Diff-LM to use employ BERT-base with 80M parameters as the backbone model and utilize a square-root noise schedule along with a diffusion forward step of 2000 and decoding steps of 200. Additionally, the embedding dimension for our models is set to 128. We use a sequence length of 128 for sentiment-conditioned generation and 256 for long-form text completion tasks.

As reported in Diff-LM, it requires approximately 5 hours to execute 200,000 iterations when trained on the E2E dataset. However, when training with the larger ROCStories dataset, which contains 98,000 five-sentence stories, it has been suggested that the algorithm be trained for at least 800,000 iterations, which requires over 20 hours of GPU time. Notably, the C4 subset contains 372.4 times more documents than ROCStories, even when the document size is not considered. As a result, at least 7,448 GPU hours would be required to adequately train the algorithm for 800,000 iterations using C4.

The official implementation of Diff-LM employs a fixed-length decoder that contains some special tokens, including paddings. As a result, it produces high Rep-4 scores. To ensure a more objective

evaluation, we performed additional post-processing to eliminate paddings and recomputed the scores based on post-processed generation.

For Genie (Lin et al., 2022), we used their official implementation as well as their fine-tuned checkpoints for XSum and CNN/DailyMail datasets, which are released officially as per Lin et al. (2022). These checkpoints are optimized using a 6-layer transformer as the encoder, pre-trained on a large 160G corpus for 5 million steps. Furthermore, a 6-layer cross-attention transformer is employed for denoising. Additionally, the latent variable dimension is set to 768, while the embedding dimension is set to 128. Genie's configuration also includes a uniform time schedule with 2,000 as the diffusion steps.

### F.4 Ablations on DDPM, diffusion steps and noise scheduler

We present ablations on DDPM vs DDIM, and PLANNER using different diffusion steps and different noise scheduler in Tab. 14. DDIM is better than DDPM across all of our experiments in most of the metrics except a slight drop in terms of diversity. We observed that more steps will typically improve the diversity score at a cost of relevance and inference speed. We also compared the cosine scheduler with the beta linear scheduler (Rombach et al., 2022). The cosine scheduler worked better in our experiments. For the summarization tasks, we found that using a noise shift Hoogeboom et al. (2023) of 4 improves the Rouge-L by around 5%, comparing to a vanilla setting with noise shift of 1.

## G Text completion with repetitive prompt

We present examples in Table 15 of generation with an ill-composed prompt for hotel review generation. The results reveal that the FT baselines tend to generate repetitive text. Although sampling mitigates this issue to some extent, self-reinforcement of repetition still occurs during generation. In contrast, our model exhibits significantly less repetitive generation.

## H Human evaluation

We screen the judges using 10 random screening questions, the judges pass 80% can participate our evaluation. The interrater agreement assessed by Krippendorff's alpha is 0.5958. The template used for human evaluation is provided in Figure 7.

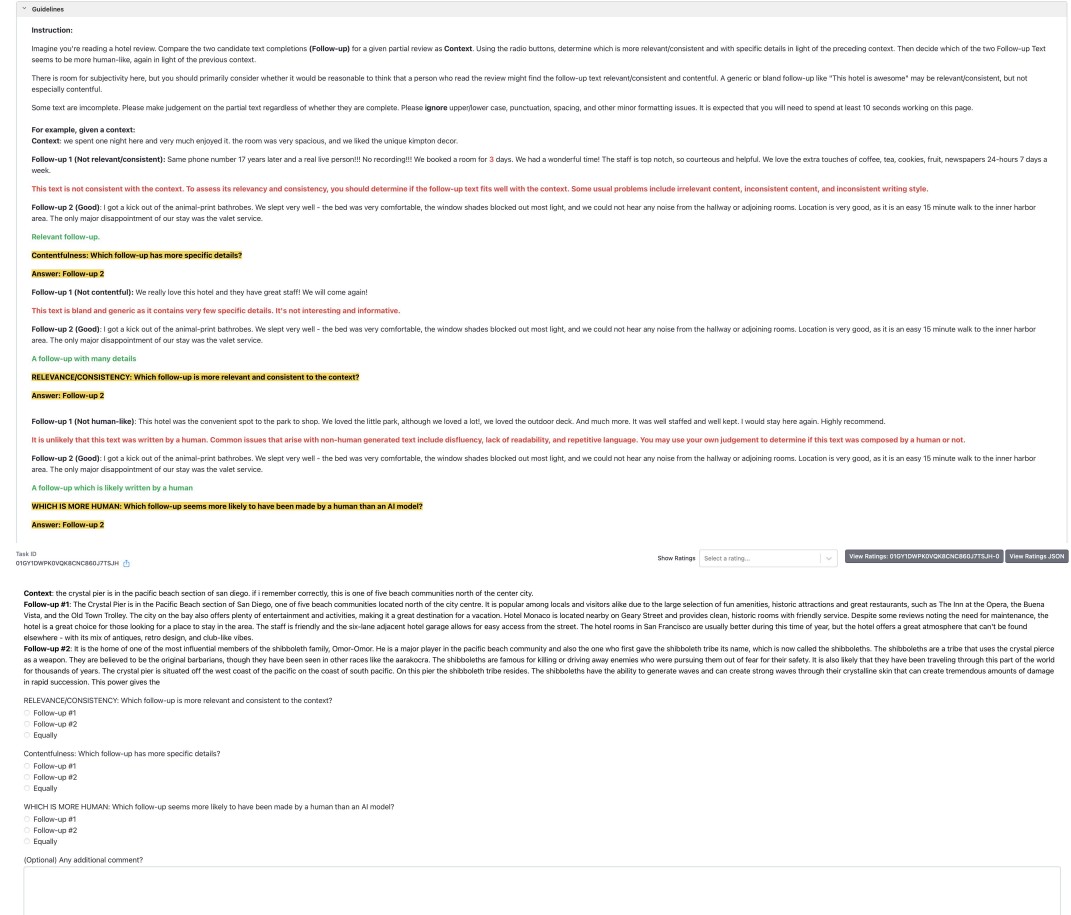

Figure 7: Template for human evaluation.

| | |
|---|---|
| Document | Washington (CNN)Maryland authorities said Wednesday that a former state correctional officer has been arrested in connection with a recent spate of shootings, including one on the Intercounty Connector in Maryland and one at Fort Meade, where the National Security Agency office is located. Officers stopped Hong Young, 35, of Beltsville, Maryland, at around 10:30 p.m. Tuesday. The officers recognized Hong's vehicle – a 1999 Lincoln Town Car – as matching authorities' description of a car seen in surveillance footage near some of the shootings. A gun in the car matched evidence found at the shootings, authorities said at a press conference, and Young was arrested. Young is in the hospital and under police guard, though when reporters asked why he was being treated, officials would only say he was arrested without incident. He is charged with attempted first-degree murder, first- and second-degree assault, weapons violations and reckless endangerment. Young worked as a correctional officer at a Jessup facility from 2012 until his resignation in 2014, Maryland Secretary of Public Safety Stephen Moyer said. There was nothing significant in his employee file, Moyer said. Police said that there are no links to terrorism, and no motive has been determined. No one was killed in the five shooting incidents, four of which occurred Monday and Tuesday, according to police reports. – February 24 in Hanover, Maryland. a man who had stopped at a Costco said a man pulled up beside him in a Lincoln Town Car at 7:30 a.m. and began firing at him. The victim's vehicle was hit several times and the victim was grazed. The assailant drove away. – March 2 in Laurel, Maryland. Police received a call at 2:50 a.m. that shots had been fired at a Walmart. There were no damages or injuries. – March 2 in Columbia, Maryland. A call came in to law enforcement at 4:51 a.m.a bout shots fired at a movie theater at Columbia Mall. Surveillance footage captured a Lincoln Town Car at about the same time shots were fired, police said. Though several employees were there, no one was hurt, authorities said. There were bullet holes in the theater glass and a shell casing was found at the scene. – March 3 in Prince George's County. Multiple shots were fired at an overpass on the InterCounty Connector in the afternoon, striking a tree service truck with two passengers inside. – March 3 at Fort Meade. Shots struck a building near the NSA office at about 6 p.m. Along with the gun, evidence shows Young was the shooter in all but the Walmart incident, though that investigation is continuing, police said. Though no one was killed in the incidents, they stirred memories of the deadly Washington, D.C.-area sniper attacks in 2002. Ten people were killed in Washington, Maryland and Virginia during that rampage, which went on for three weeks. CNN's Holly Yan and Laurie Ure contributed to this report. |
| Time 0.95 | Police: man in "gunman" shooting in police shooting, police say. Police in a car, police in a vehicle, police say. |
| Time 0.90 | Police: man are in police shooting, murder in police shooting, police say. Police in a vehicle found in police shooting, police say. |
| Time 0.85 | Police: man arrested in one murder, one in shooting, police say. Police find a car, a police vehicle in the suspect, police say. |
| Time 0.80 | Police: Young man arrested in attempted murder, one in shooting, police say. Officers in a car matching the vehicle found in surveillance footage, police say. |
| Time 0.75 | Police arrest man, man charged with attempted murder, shootings, police say. Officers say a car matching police description found in surveillance images, police reports. |
| Time 0.70 | Police arrested Lee Young, with attempted murder, assault charges, weapons violations, authorities say. Officers found a car matching surveillance footage found in the vehicles, reports say. One of the shootings, no one was in the car. |
| Time 0.65 | Police arrested Hong Lee, 35, with attempted murder shootings, weapons violations, authorities say. Vehicle in his car matched surveillance footage found in surveillance images, police say. Only one in five shootings were no one was killed in Maryland, police say. |
| Time 0.60 | Police arrested Hong Young Lee, 35, with attempted first-degree murder, authorities say. Car in the car matched surveillance footage found in surveillance images, police say. One of the five shootings occurred in the home. No one was killed in Maryland, but no motive has been determined. |
| Time 0.55 | Police arrested Hong Young Lee, 35, with attempted first-degree murder, authorities say. Car in the car matched the surveillance images found in surveillance footage, police say. No one was killed in the five shootings. |
| Time 0.50 | Police arrested Hong Young, 35, for attempted first-degree murder, authorities say. Car in the car matched the evidence found in surveillance footage, police say. |
| Time 0.45 | Police arrested Hong Young, 35, for attempted first-degree murder, authorities say. Car in the car matched the evidence found in surveillance footage, police say. |
| Time 0.40 | Police arrest Hong Young, 35, in attempted first-degree murder, authorities say. A car matched surveillance images found in surveillance footage. |
| Time 0.35 | Police arrested Hong Young Li, 35, in attempted first-degree murder, authorities say. A car in the car matched surveillance images found in surveillance footage. |
| Time 0.30 | Police arrested Hong Young Li, with two attempted shootings, assault charges, authorities say. A gun in the car matched surveillance images found in surveillance footage. |
| Time 0.25 | Police arrested Hong Young, with two attempted shootings, assault charges, authorities say. A gun in the car matched surveillance images found in surveillance footage. |
| Time 0.20 | Police arrested Hong Young, with two attempted shootings, assault, authorities say. A gun in the car matched surveillance images found in surveillance footage. |
| Time 0.15 | Police arrested Hong Young, 35, with attempted first-degree murder, assault, authorities say. A gun in the car matched surveillance images found in surveillance footage, police say. No one was killed in the five shootings. No motive has been determined. |
| Time 0.10 | Police arrested Hong Young with attempted first-degree murder, assault, authorities say. A gun in the car matched forensic identification found in surveillance footage, police say. No one was killed in the five shootings. Weapons violation. No motive has been determined. |
| Time 0.05 | Police arrested Hong Young with attempted first-degree murder, assault, authorities say. A gun in the car matched forensic DNA found in surveillance footage, court documents show. No one was killed in the five shootings. Weapons violations. No motive has been determined. |
| Time 0.00 | Police arrested Hong Young after attempted shootings, assault, authorities say. A gun in the car matched forensic DNA found in surveillance footage, police say. No one was killed in the shootings in Maryland. Weapons violations, police say. No motive has been determined. |

Table 10: Generation from the diffusion model with 20 steps on CNNDM dataset. In last two steps, the highlighted text regarding "forensic DNA" is hallucinated, while the previous generations (e.g. " A gun in the car matched surveillance images") are more reasonable.

| Method | PPL | DIST/ENT | S-BL | Rep-4 | BLEU | ROUGE | Len |
|---|---|---|---|---|---|---|---|
| PLANNER greedy | 47.3 | 0.17/6.60 | 0.52 | 1.55% | 0.77 | 7.9 | 168.1 |
| PLANNER top-p | 72.0 | 0.20/6.80 | 0.38 | 0.94% | 0.58 | 6.1 | 173.2 |

Table 11: PLANNER with stochastic decoding yields higher diversity at a cost of other metrics.



| | Semantic generation (*hotel review*) |
|---|---|
| PLANNER (Negative) | I've received several complaints about the amenities of hotels, although the rooms are standard La Quinta, and the price is a bit low. I booked a room for a family of four. Given the low price, I expected to tolerate the motel's unappealing colors, but that's probably par for the course. I read the reviews and chose this hotel for a night sleep, but I was incredibly disappointed, to say the least. Part of the issue might have been the noise from the freeway, compounded by the fact that I could hear the AC from my neighbor's room. There was also a noise that sounded like "flooring". The worst part was that the bed sheets had no cover, and the AC draft was felt throughout the room. The room itself was cramped, slightly outdated, and uncomfortable. The furniture was old, but there wasn't much else in terms of extras. If it weren't for the free drinks and the front desk service, I would have rated this hotel much lower than most others. |
| PLANNER (Positive) | I've got numerous praises about the amenities of the hotel. Rooms are typical of La Quinta's excellent standards, and the price is low surprisingly. I booked a room from they website. I was surprised for the hotel's distinctive color scheme, quite a signature style. This hotel I selected for overnight stay, and I was overwhelmingly satisfied with no complaints. A key element of the charm was the profound tranquility throughout the hotel. Despite being conveniently near the freeway, the excellent soundproofing ensures a peaceful stay. Which was great was that the bed sheets were so soft and comfortable, and the gentle breeze from the AC added to the overall soothing ambiance. The room itself was cozy, with a charming vintage vibe, and supremely comfortable. The furniture was classic, giving a nostalgic touch. The complimentary drinks and the outstanding service at the front desk were delightful, making this hotel a great choice. |
| Retrieved from training set | The hotel is located in a great area, close to a lot of things. After reading a lot of the previous reviews I booked it and decided to see for myself. The checkin was a little frantic but there were a lot of people so I wouldn't count that against them. The gentleman that checked me in was extremely polite and apologized several times for the wait, which hadn't been that long. I went up to my room on the second floor and was surprised that it was bigger than I thought it would be. I did have the tiny bathroom that previous posters commented on but it was fine. My only complaint would be the noise level. I was in a room that faced the street and the noise was unbelievably loud. If I stayed here again I would ask for a room which doesn't face the street. It was nice to have a Starbucks right off the lobby, as well as a coffee maker in the room. So, for the money, I would recommend the Hotel Diva. |
| | Text completion (*C4 subset*) |
| Context | Operating a successful dental practice isn't easy. That's why we've used our industry knowledge and expertise in practice management to create an eBook that offers tips on how you can create and maintain a financially healthy and profitable dental operation. |
| PLANNER | This e-book is akin to a popular one, if not more stylish. It provides all the benefits you need for your accounting endeavors for your dental practice. Unlike other books, this practice management guide will enlighten you on what it's like to pay bills and kickstart your business. You'll learn how to effectively stage your practice management, and the book offers valuable tips and financial ideas for launching your dental business. It delves into a fascinating topic, presenting the steps you need to enhance your financial productivity and support your financial growth plan. You'll discover how to manage your ongoing fiscal and accounting needs, and receive tips that will enable you to achieve timely results and make informed financial decisions. The book includes quick and detailed insights on driving numbers and finding solutions. Learn how to start your accounting practice and achieve the ultimate goal of creating a profitable and successful business. We appreciate your feedback! Leave a review, and we'll send a book for you. We offer examples of the key things you should consider when deciding to start your dental business. |
| | Summarization (*CNN-Dailymail*) |
| Document | (CNN)Actress Liza Minnelli has entered a rehab facility for her addictions, a spokesman said. "Liza Minnelli has valiantly battled substance abuse over the years and whenever she has needed to seek treatment she has done so," said spokesman Scott Gorenstein. "She is currently making excellent progress at an undisclosed facility." The 68-year-old has struggled with addictions to alcohol and painkillers in the past. Minnelli won an Oscar in 1973 for her performance in "Cabaret." CNN's Topher Gauk-Roger contributed to this report . |
| PLANNER | The actress Liza Minnelli is "making excellent progress" at a undisclosed facility, spokesman says. She won an Oscar in 1973 in "Cabaret" |
| | Summarization (*XSum*) |
| Document | Japanese exports rose by just 0.6% in September from a year earlier, much slower than in previous months. The data is seen as reflecting a slowdown in Japan's top trading partner, China. However, the Nikkei 225 closed 1.9% higher at 18,554.28 on hopes the weak data would trigger more stimulus efforts from the government. In China, the mainland's benchmark Shanghai Composite closed down 3.06% at 3,320.68. Hong Kong's Hang Seng remained closed on Wednesday for a public holiday. In Australia, the S&P/ASX 200 rose by 0.2%, closing at 5,248.30. Shares in mining giant BHP Billiton rose 1% after it maintained its full-year production guidance for iron ore. Slowing growth in China has been weighing on mining stocks in recent weeks as the country is the main importer of Australian commodities. South Korea's benchmark Kospi index rose marginally by 0.2% to finish at 2,042.98. Shares of LG Electronics jumped by more than 14%, boosted by a fresh supply agreement with General Motors. |
| PLANNER | Asian markets closed higher after weak economic data indicated slow growth due to lower exports. |

Table 12: Generated examples.

| | Summarization (***CNN-Dailymail***), hallucination issue | |
|---|---|

| Document | (CNN)A Los Angeles police officer charged with murder in the shooting death of a local truck driver had his father's help in fleeing California and traveling to Texas, court documents filed Thursday said. Henry Solis, a Los Angeles Police Department rookie, is accused of killing 23-year-old Salome Rodriguez Jr. early in the morning of March 13 in Pomona. Solis disappeared shortly after the shooting. Items left by Solis at the crime scene allowed Pomona police detectives to identify him "as the individual responsible for killing Rodriguez," according to an affidavit signed Thursday by FBI Special Agent Scott Garriola. The document goes on to say that Solis made "incriminating statements about his role in the murder" to friends, family members and others, and said he would never be seen again. The affidavit says his father, Victor Solis, quickly left his home in Lancaster after receiving a call from him. The elder Solis was seen a day later in El Paso, Texas, at the home of family members. He told FBI agents he drove his son to El Paso, dropped him off at a bus station and doesn't know where he is. Federal authorities haven't said if Victor Solis would face charges for helping his son leave California. CNN's Tina Burnside contributed to this report. |
|---|---|
| PLANNER | Henry Solis is accused of killing Delara Salome Guzman Jr., Solis disappeared shortly after the shooting. Victor Solis helped his son leave California. |

| Document | Two men have died after being crushed by timber that fell on top of them inside a shipping container. Authorities were called to an industrial site in Wetherill Park, west of Sydney, shortly after 11am on Thursday where they worked to release three men. Police found the three men inside a shipping container on a property on Victoria Street. Authorities were called to an industrial site in Wetherill Park, west of Sydney, shortly after 11am on Thursday where they worked to release three men . They had been unloading or loading timber when it fell on them. Two men died at the scene as a result and the surviving man was taken to nearby Fairfield Hospital in a serious condition. Police say the shelving in the shipping container moved and trapped the men. Police and WorkCover NSW are investigating the incident and a report will be prepared for the Coroner. Two men died at the scene as a result and the surviving man was taken to nearby Fairfield Hospital in a serious condition . |
|---|---|
| PLANNER | Authorities were called to Worsley Park, west of Sydney, Sydney. Police found three men inside a shipping container after 11am on Thursday. Two men died at the scene as a surviving man was taken to nearby hospital. NSW police are investigating the incident. |

| | Text completion (***C4***), fluency issue | |
|---|---|

| Prefix | Can't quite find what you're looking for? Try our handy site search. |
|---|---|
| PLANNER | Fill our search engine form and try us and we'll try to get back to your desk on time. We have a reliable staff that provides professional service, whatever you require - not just a specific search function. But we also let's you enjoy on-line selection of different products, storage, protection, tools, books and accessories, and many more products. Simply what you need! Our main site is the updated web site site you can visit. We don't store any coupons, promotions, products and printed materials for animals and get them for products or items in the shopping cart that must always be done by a local team of technicians. Including customer information. It is important to the customers by checking their address information. Or if you mean on the help page, we'll get it. |

Table 13: Problematic examples.

| Arch. | PPL | DIST/ENT↑ | S-BL↓ | Rep-4↓ | BL↑ | R-L↑ | Score↑ | Len |
|---|---|---|---|---|---|---|---|---|
| | | | DDIM vs DDPM | | | | | |
| DDIM | 47.36 | 0.17/6.602 | 0.52 | 1.55% | 0.77 | 7.9 | 0.55 | 168.08 |
| DDPM | 57.34 | 0.18/6.663 | 0.44 | 1.48% | 0.35 | 5.7 | 0.53 | 162.81 |
| | | | Different Inference Steps | | | | | |
| 5 steps | 53.215 | 0.17/6.547 | 0.54 | 1.81% | 0.67 | 7.2 | 0.55 | 134.2 |
| 10 steps | 47.807 | 0.17/6.580 | 0.5 | 1.60% | 0.69 | 7.4 | 0.55 | 138.78 |
| 20 steps | 47.559 | 0.17/6.581 | 0.52 | 1.57% | 0.71 | 7.7 | 0.55 | 146.38 |
| 30 steps | 47.36 | 0.17/6.602 | 0.52 | 1.55% | 0.77 | 7.9 | 0.55 | 168.08 |
| 50 steps | 47.096 | 0.17/6.605 | 0.54 | 1.56% | 0.83 | 7.9 | 0.55 | 162.08 |
| | | | Scheduler | | | | | |
| Cosine | 47.36 | 0.17/6.602 | 0.52 | 1.55% | 0.77 | 7.9 | 0.55 | 168.08 |
| Beta Linear | 49.78 | 0.17/6.577 | 0.57 | 1.46% | 0.64 | 6.8 | 0.53 | 158.83 |

Table 14: Ablations on hotel review completion.

| | |
|---|---|
| Prompt | This hotel is a beautiful historic building. Our room was huge. This hotel is really a beautiful historic building. |
| FT-search | The hotel is located in the heart of the city, right next to the historic center of the city. The hotel is located in the heart of the city, right next to the historic center of the city. The hotel is located in the heart of the city, right next to the historic center of the city. The hotel is located in the heart of the city, right next to the historic center of the city. The hotel is located in the heart of the city, right next to the historic center of the city. The hotel is located in the heart of the city, right next to the historic center of the city. The hotel is located in the heart of the city, right next to the historic center of the city. The hotel is located in the heart of the city, right next to the historic center of the city. The hotel is located in the heart of the city, right next to the historic center of the city. The hotel is located in the heart of the city, right next to the historic center of the city. The hotel is located in the heart of the city, right |
| FT-sample (sample 1) | our room was huge. this hotel is really a beautiful historic building. our room was huge. this hotel is really a beautiful historic building. our room was huge. this hotel is really a beautiful historic building. our room was huge. this hotel is really a beautiful historic building. our room was huge. this hotel is really a beautiful historic building. our room was huge. this hotel is really a beautiful historic building. our room was huge. this hotel is really a beautiful historic building. our room was huge. this hotel is really a beautiful historic building. our room was huge. this hotel is really a beautiful historic building. our room was huge. this hotel is really a beautiful historic building. our room was huge. this hotel is really a beautiful historic building. our room was huge. this hotel is really a beautiful historic building. our room was huge. this hotel is really a beautiful historic building. our room was huge. this hotel is really a beautiful historic building. our room was huge. this hotel is really a |
| FT-sample (sample 2) | we stayed here in the early morning and it was amazing. the staff is very nice and helpful and the room was very spacious. we are now staying in the city of cedarhill. it was a beautiful place to stay and it is a very nice hotel. we are currently going to stay here in the city of cedarhill. I have lived in the city of cedarhill since 2014. It has definitely transformed the look of the city. The city is now extremely walkable for work and recreation. I am from Colorado so my city is more suburban and rural than it was five years ago. That is a plus. But cedarhill still has that unique, historic feel of it's past and it is definitely growing in size and scope. We have not had a problem with any of our pets and we have been welcomed with open arms. I absolutely recommend cedarhill. As a person who has lived and visited both Fort Collins and Cedar Rapids, my family's travel plans have changed dramatically as the cities both continue to grow. Cedar Rapids has become a more important city in our minds and we want to make it a memorable experience |
| PLANNER (sample 1) | The room was well appointed (amazingly comfortable) and we had spacious bathrooms (which were great!) and the $99 rate was great!!! The staff was exceptional at the very least, and we had got there on 3 nights so we checked in there early! We walked from the Riverwalk by the Seattle Convention Center, as the restaurants are very close to hotels. Many of the things are on the road and take a bus to get downtown. The staff at the hotel were all helpful and friendly, and hotel is close to the shopping/dining district. They are literally right across the street.....just blocks away from the hotel! |
| PLANNER (sample 2) | On the other hand, not huge kitchenette but really equipped with everything one needs. As the staff, all were very helpful. The restaurant in the hotel is very very nice area of the restaurant, so you can enjoy the main area of the restaurant although we ordered on hand food and it was a large buffet. The only problem I have was how the air con was either too loud or was on the cold side which was difficult but it seems it really helps to drown out the noise. I will definitely recomend their hotel restaurant... The food was devine with service and of food quality. I just cant wait to experience the hotel restaurant in time to have a great meal at the bar and in the main lobby for a drink in the morning. The bar is so nice, coming in for its so historic atmosphere so you can see how people could tell they were a lot of history. |

Table 15: Generation with repetitive prompt.