# OpenReview forum: "PLANNER: Generating Diversified Paragraph via Latent Language Diffusion Model"
_NeurIPS.cc/2023/Conference — NeurIPS 2023 poster_

### Official Review · Reviewer_H5ac · 2023-07-05

**Soundness:** 3 good
**Presentation:** 4 excellent
**Contribution:** 3 good
**Rating:** 7
**Confidence:** 4

**Summary:**

The paper presents a two-stage latent text diffusion model that uses an autoencoder to condense lengthy texts into a limited number of paragraph embeddings, and a continuous time diffusion model that359 learns the distribution of these embeddings. The paper presents detailed experiments on open-ended generation tasks and shows that the proposed model alleviates the issue of repetition and advances generation diversity across different tasks.

**Strengths:**

The paper is well-motivated. The examples in Figure 1 with the self-reinforcing effect are impressive. The proposed method first summarizes the paragraph information with an autoencoder and then applies the diffusion process to control the token generation process, which is novel from my point. The paper conducts extensive experiments on open-ended generation tasks and provides some analyses of the running time.

**Weaknesses:**

1. Could you provide more experimental analyses on the learned paragraph embedding? I think that paragraph embedding is used to learn a high-level concept for planning but ignores the details of sentence/phrase structure to avoid copying similar phrases from previous prompts and repeating them.  Could you provide more experiments to show the quality of learned paragraph embedding? Directly extracting the first K tokens from the encoder is not the best choice for me.

2. In experiments such as Table 1, the proposed method with a greedy decoding mode is comparable to baselines with top-p sampling in terms of diversity. A question is whether PLANNER combined with top-p can lead to better results. Are the proposed methods compatible with stochastic decoding methods?

**Questions:**

Please see the weakness.

**Limitations:**

Please see the weakness.

---

> ### Author Rebuttal · Authors · 2023-08-09
>
> We thank the reviewer for their encouraging feedback. We address the questions in below:
>
> *1. Paragraph Embedding*
>
> Please refer to the general question titled "Motivation and Impact of $k$." Additionally, we provide an analysis of how the acquired embedding influences the eventual generation, along with examples illustrating the abilities of reconstruction, denoising, and interpolation in Appendix A.
>
> *2. PLANNER with stochastic decoding*
>
> It is possible and straightforward to implement stochastic decoding for the PLANNER. In fact we experimented with this option. In our experiments, we utilized nucleus sampling with a value of p=0.92 and K=50 (see below for an experiment on hotel review generation). By incorporating stochastic decoding, the diversity and repetition metrics can be improved, although at the expense of relevance and accuracy scores. It is important to mention that the decoder's role in PLANNER is to faithfully translate the latent code into the desired target text, rather than performing compositional/planning job. Stochastic decoding may disrupt this role and can lead to undesirable generation, as we observed an increase in hallucinations when combining PLANNER with stochastic decoding.
>
> | Method          | PPL  | DIST/ENT   | S-BL | Rep-4    | BLEU  | ROUGE | Len   |
> |-----------------|------|------------|------|----------|-------|-------|-------|
> | PLANNER greedy  | 47.3 | 0.17/6.60  | 0.52 | 1.55%    | 0.77  | 7.9   | 168.1 |
> | PLANNER top-p   | 72.0 | 0.20/6.80  | 0.38 | 0.94%    | 0.58  | 6.1   | 173.2 |

---

### Official Review · Reviewer_6a1h · 2023-07-06

**Soundness:** 2 fair
**Presentation:** 3 good
**Contribution:** 3 good
**Rating:** 6
**Confidence:** 4

**Summary:**

This paper presents a 2-stage generative model for text. The first stage involves training a VAE over the data to obtain an effective encoder and decoder modules. The second stage uses the output of the frozen VAE encoder as the hidden state of the input text and learns a regular continuous diffusion model over the encoded hidden states of training data instances. For generation, the diffusion model produces a hidden state from noise and this hidden state is fed into the VAE decoder to generate text. This model is evaluated on sentiment-guided generation, text completion, and summarization and compared against baselines that include token-based diffusion models (instead of hidden state text diffusion models in this work), VAE/Enc-dec models, and standard autoregressive models (like fine tuned GPT-2).

**Strengths:**

– Although the approach seems straightforward, to the best of my knowledge, surprisingly I am not aware of other latent sequence/paragraph embedding based diffusion models for text generation. Hence, this fills a gap in the research on diffusion models for text.

– The experimental design is reasonable and informative.

– This approach seems to be more fluent, diverse, and effective than popular token-based diffusion models following the results.

– Although typically less fluent than the autoregressive models, it still performs competitively when considering other metrics like diversity, reference overlap, and control.


**Weaknesses:**

– I have some concerns about evaluation, particularly related to the quality of baseline models. For example, on CNN-DM summarization task, publicly available results (https://paperswithcode.com/sota/abstractive-text-summarization-on-cnn-daily) show that T5 achieve R-L that is ~7-10 points better than reported in this paper (and is hence better than the R-L proposed model achieves). Therefore, I am doubtful about the quality of training/tuning of the baseline models reported in the paper.

– Choosing “first k hidden state vectors” from the encoder in the VAE seems arbitrary. What is the motivation behind this hyperparameter? What is the effect of this k?

– The paper alludes to learning “paragraph embeddings” which capture high level semantic properties which enables “coarse-to-fine” generation and allows for “planning”. However, no evidence is presented in this paper about compositional/planning capabilities of the model. While it is true that paragrpahs/tokens are generated from a lower dimensional manifold, it doesn’t automatically imply that these embeddings are linguistically interesting or allow for controlled planning-oriented generation. At an abstract conceptual level, how different are these representations from VAE based representation for example?

– Although this is a generative model, the training objective is arbitrary and doesn’t seem linked to maximizing the likelihood/ learning the distribution of the training data. This is because the VAE and the diffusion components are separately trained and hence preclude a clean interpretation of the training objective.

– I am not convinced that the “Distributional smoothness metric” reflect the nature of the manifold as is claimed in the paper.

– Runtime analysis seems to compare unbatched autoregressive models with batched diffusion models. Sorting by length and then padding for batching is a standard practice for autoregressive models.


**Questions:**

See above.

**Limitations:**

See above.

---

> ### Author Rebuttal · Authors · 2023-08-09
>
> We thank the reviewer for their insightful feedback. Next, we address the comments:
>
> *1. Weak baseline*
>
> Thank you for the pointer. We have examined the link. Please correct us if we are making mistake, but it seems to us that the T5 model that achieved 7-10 points better than our T5 baseline might be much larger (11B, [https://paperswithcode.com/paper/exploring-the-limits-of-transfer-learning](https://paperswithcode.com/paper/exploring-the-limits-of-transfer-learning)) than our baseline (770M). We also found some publicly accessible model ([https://huggingface.co/sysresearch101/t5-large-finetuned-xsum-cnn](https://huggingface.co/sysresearch101/t5-large-finetuned-xsum-cnn)) that have finetuned T5-large using XSum + CNNDM, and they have reported lower ROUGE scores than ours.
>
> *2. Motivation of $k$*
>
> Please refer to general questions.
>
> *3. Planning*
>
> Thank you for the insightful comment. According to our understanding, the diffusion model engages by utilizing the previous step output and the controlling signal to refine the representation and incorporate more intricate details. For an example, please refer to Appendix Table 8. Initially, the decoded text from early diffusion steps lacks specificity and coherence. However, as the diffusion inference progresses, the model gradually incorporates additional syntactic and semantic details, and also determines when to remove certain information. This process bears resemblance to the image diffusion process. The external controlling signal provides guidance at each stage of the text generation process. While the representation is learned by VAE, our approach differs in that the diffusion model generate samples close to the ones from the posterior distribution of the latent code, which potentially possesses a more sophisticated structure than a simple Gaussian distribution.
>
> *4. Arbitrary objective*
>
> Our understanding is that Diffusion is the process of optimizing the evidence lower bound (ELBO) as VAE, which is a valid objective. Our diffusion model is essentially a Latent diffusion model (LDM) which share the same objective thus remains meaningful.
>     Our two stage training setup resembles that of VQVAE/VQVAE2 (which actually reports likelihoods). In VQVAE, they state "Whilst training the VQ-VAE, the prior is kept constant and uniform. After training, we fit an autoregressive distribution over z, p(z), so that we can generate x via ancestral sampling." We share similar rationale with VQVAE. In the first phase, we train a regular VAE with a simple prior and posterior; In the second phase, we freeze the decoder and posterior, but train a diffusion-model-based prior, which only optimizes the $KL(q(z|x) || p(z))$ term but freezes the $p(x|z)$ part. As a result, the ELBO should continue to improve.
>
> *5. Distribution smoothness metric*
>
> We acknowledge that the PPL of the linear-interpolated sample might not fully reflect the nature of the manifold. Nevertheless, we would hope it has the potential to provide some insight into whether the posterior distribution is highly multimodal, with spikes and numerous density "holes". Given our knowledge we might not be aware of other better alternatives to evaluate the smoothness of a distribution. Therefore, we would greatly appreciate any suggestions that could expand our understanding.
>
> *6. Running time analysis*
>
> We are aware that the unbatched autoregressive model is not the best choice (see line 343). When we evaluate the full test set of CNNDM and XSum we performed the batched version by sorting input text by length and maximally batchifying them as possible. The total running time reduced more than 4x from 11 hours (unbatched version) to 2.5 hours (batched version). We admit there might be more room to accelerate the autoregressive baseline. In fact, we do not intend to indicate our method can be faster than the autoregressive ones, but only to show the convenience of arranging input into the same length vectors. We will make this clearer.

---

> > ### Comment · Reviewer_6a1h · 2023-08-16
> > **Thanks for the response**
> >
> > I am willing to agree that the results I mentioned might have been achieved by a much larger model. However, my overall impression remains similar as before so I am keeping my current score.

---

### Official Review · Reviewer_r7NC · 2023-07-06

**Soundness:** 2 fair
**Presentation:** 3 good
**Contribution:** 2 fair
**Rating:** 5
**Confidence:** 4

**Summary:**

In this paper, the authors propose PLANNER, a model that combines latent semantic diffusion with autoregressive generation to generate fluent text while exercising global control over paragraphs. The proposed method is evaluated on various conditional generation tasks, and the results show its effectiveness in generating high-quality text.

**Strengths:**

1. The paper addresses the issue of repetitive and low-quality output generated by autoregressive models and proposes a novel approach using latent semantic diffusion.
2. The combination of autoregressive decoding and latent diffusion allows for efficient generation over paragraph generation.
3. The proposed method is evaluated on various tasks and shows improved generation quality compared to autoregressive and text diffusion baselines.

**Weaknesses:**

1. I understand that revisiting and revising the generated sentences can alleviate exposure bias, as errors can be reduced through further editing at the token level. However, revisiting and revising the latent space does not seem reasonable to me. From my perspective, the exposure bias occurs in the process of picking words out, but the authors merely employ GPT-2 without making any changes in the phase.
2. The paper claims that they can generate longer text and paragraphs, but there is no further analysis about relationship between length of generated sentence and the performance.
3. The evaluation of CNN/Daily Mail and XSum is performed on 256 subsampled examples from the test set. This manner is not convenient for the subsequent works to follow, and this also hinders reviewers from making fair comparisons between this work and others.
4. I have reservations about the ability of the variational paragraph embedder to learn effective representations of sentences with different lengths. What would happen if there is a significant difference in sentence lengths, such as one being very short (e.g., 5 tokens) and the other being much longer (e.g., 512 tokens or more)?

**Questions:**

1. Unlike autoregressive models based on conditional probabilities, diffusion models are unable to ascertain the optimal sentence from the generated set. How did you address this problem in your paper?
2. As the reverse diffusion goes on in latent space, does the latent representation z get closer to the representation of the ground-truth one in cosine distance or other distances?
3. What is the influence of paragraph embeddings number, k in line 119?

**Limitations:**

---

> ### Author Rebuttal · Authors · 2023-08-09
>
> We thank the reviewer for their constructive feedback. We address the comments in below.
>
> *1. Evaluate on subset datasets*
>
> Please refer to general questions.
>
> *2. Exposure bias*
>
> Our hypothesis is that exposure bias occurs when there is a discrepancy between the training and inference stages, specifically during teacher forcing. In our diffusion model, we predict the latent semantics code in an non-autoregressive manner. No partial ground truth latent code has been fed into the diffusion model during training time. Consequently, the diffusion model is less affected by exposure bias issue.
> As for the decoder, it is trained using an autoencoder with teacher forcing, which means that exposure bias can still exists in this stage. However, the impact of exposure bias is limited to the final translation of the semantics into text. We demonstrate that the error in this translation is minimal (reconstruction BLEU > 80\%) due to the simplicity of the autoencoding task and the strong influence of the input latent code on the decoder, resulting in less error compounding effects. Hence, we experience much less exposure bias compared to the autoregressive approach.
>
>
> *3. Length Ablation*
>
> We conducted tests on a total of five datasets using our model. The target generation length varied within a range of 15.2 to 181.29. This range was selected to encompass diverse generation lengths. If comparing across tasks, our method received more diversity/repetition metrics improvement in lengthier generation tasks (review generation).
> In the hotel reviews dataset, the length of the target sentences can vary from 23 to 512. In Appendix Table 6-7, we provide examples of sentences reconstructed from the latent codes. Generally, shorter sentences are easier to represent and reconstruct, evidenced by the fact that tasks with shorter target sentences (XSum, CNN-DM) achieved higher $BLEU_{clean}$ scores. We will make these clearer.
>
> *4. No optimal solution*
>
> Our method, inheriting from the latent diffusion model, is a sampling technique. It is worth noting that even for tasks such as translation and summarization, the default generation methods for autoregressive models like popular LLMs rely on sampling. Previous work [1,2] has demonstrated that the notion of an "optimal sentence" may be a misleading "red herring", as optimizing the likelihood can result in low-quality outputs such as repetition and generation artifacts. This is particularly evident in open-ended generation scenarios, where the distribution of text is inherently multimodal.
>
> *5. Representation Distance*
> Yes, as the reverse diffusion process progress, the representation will be closer to the ground truth representation. This is evidenced by the Figure 6 in the Appendix, which shows the BLEU score between the ground truth text and the text translated from the latent code at time $t$. The graph reveals a consistent pattern of progressive improvement in the BLEU score as $t$ decreases from 1 to 0.
>
> *6. Impact of $K$*
>
> Please refer to general questions.
>
> [[1]](https://arxiv.org/abs/1904.09751) Ari Holtzman, Jan Buys, Li Du, Maxwell Forbes, and Yejin Choi. *The curious case of neural text degeneration.* In ICLR, 2019
>
> [[2]](https://arxiv.org/abs/2206.02369) Jin Xu, Xiaojiang Liu, Jianhao Yan, Deng Cai, Huayang Li, and Jian Li. *Learning to break the loop: Analyzing and mitigating repetitions for neural text generation.* In NeurIPS, 2022

---

> > ### Comment · Reviewer_r7NC · 2023-08-18
> >
> > Thank you for the clarifications. I will raise my rating.

---

### Official Review · Reviewer_MZNE · 2023-07-15

**Soundness:** 4 excellent
**Presentation:** 3 good
**Contribution:** 4 excellent
**Rating:** 7
**Confidence:** 4

**Summary:**

This paper proposes to combine latent semantic diffusion with autoregressive generation to alleviate the issues of exposure bias in training / inference of text based language models, and computational and performance cost of purely diffusion approaches. They improve the diffusion process by applying it to the latent semantic space instead of the token / embedding space. For this, they learn some "semantic tokens" for encoding paragraph level information and then use a decoder to map these to the raw text space.

**Strengths:**

Working with the diffusion model in the semantic spaces opens up the door for controllable generation.
They also provide an extensive study of the requirements for a good latent space for paragraph diffusion models.
The paper is well written and easy to follow.
The ideas employed to fix / ensure local smoothness (by perturbing data) and distributional smoothness (by using VAE) are simple and useful.
They propose a novel (?) metric called AuBLEU to evaluate the denoising capabilities of the model. I believe is this generally suitable for other works and could be impactful. However, it does not feel to be properly justified / grounded.
Hparams are provided.
 Human eval results are significant. The other results are sound and robust evaluation is performed. The paper is just lacking ablations on the design choices.
The analysis is complete and justifies the main claims made by the authors.

**Weaknesses:**

The changes employed by the authors, especially during the training stage are not properly ablated.
It is not clear if the proposed fixes provide benefits.
Some of the design choices are not experimentally justified (eg: line 182)
Evaluation is performed on just a sample of the test set (this is the first time I have seen something like this in a paper and I'm not sure how to take it - I'm not super comfortable as this makes your technique essentially un-comparable). This also might not be robust. (line 196).
No intention of providing code and it might be very hard to reproduce because of the many changes especially in the training setup.


Nitpicks:
typo line 181


**Questions:**

Can you talk a bit more about the design choices and how much impact they had on the results? For eg: how much impact does fixing "distributional smoothness" have?
Have you performed other ablation experiments to justify claims?
Why did you subsample the test sets?

**Limitations:**

Addressed in appendix.

---

> ### Author Rebuttal · Authors · 2023-08-09
>
> We thank the reviewer for their helpful feedback. We will fix the typo pointed out. We address the questions in below:
>
> *1. Evaluate on subset datasets*
>
> Please refer to general questions.
>
> *2. Ablation regarding rescaling*
>
> We omit the rescaling step in this study due to the absence of the "rescaling invariant" property in the latent text code. Specifically, we have not imposed any constraints to ensure that the generated output remains the same after rescaling the latent code. In contrast, for Imagen, where the generation takes place in the raw pixel space, rescaling will predominantly retain the shape information while altering only the contrast and brightness. Initially, we conducted experiments involving rescaling, but results demonstrated poorer performance compared to the non-rescaled version, as evidenced by a **-10.6** ROUGE score drop on CNNDM. Consequently, we opted for the dynamic thresholding without rescaling. We will make this clearer.
>
> *3. Code Release*
>
> We have a python+pytorch implementation that reproduces experiment results, and are finalizing legal approvals to open-source the codebase. We will release the code upon publication.
>
> *4. Ablation on design choice*
>
> We indeed performed an ablation for "distribution smoothness" on the sentiment-guided generation task. The model trained without the variational objective underperforms the full model in all metrics by **-10.1** AuBLEU, **+18.8** PPL, and **-15.2%** ACC. We will incorporate the results in the next revision.
>     The evaluation pipeline might get a bit expensive if the diffusion model training and evaluation is also involved. Instead, we mostly use a surrogate metric (line 255) to monitor the overall quality of the learned representation. Detailed evidence presented in the appendix demonstrates a reasonably strong correlation between this empirical metric of the representation quality and the subsequent performance of diffusion generation. The models trained with the variational objective consistently improve the performance across the board.

---

> > ### Comment · Reviewer_MZNE · 2023-08-14
> >
> > I acknowledge the rebuttal and appreciate the full evaluation. Can you add statistical tests to your evaluation results as well?
> >
> > I thank the authors for the clarification and would like to stick with my initial score.

---

> > > ### Author Response · Authors · 2023-08-17
> > >
> > > Thank you for reading our response and your additional suggestions! We will perform the statistical analysis on the evaluation results in our next revision.

---

### Author Rebuttal · Authors · 2023-08-09

**Common Questions**:

*1. Evaluate on subset datasets*

In order to expedite the iterations of the experiment, we opted for a partial evaluation of our method as the full evaluation of our method / Genie takes 7h / 2d to complete on CNN-DM or XSum test set. Nevertheless, we agree with the reviewers' concern that this approach could potentially compromise the comparability of our results. Consequently, we have conducted a full evaluation on the entire test set of CNNDM and XSum, which is presented in the table below. The main conclusion remains the same. We will include all the updates in our forthcoming version.


| **Arch.** | **PPL** | **DIST/ENT**↑ | **S-BL**↓ | **Rep-4**↓ | **BL**↑ | **R-L**↑ | **Score**↑ | **Len** | **AuBL**↑ |
|:---------:|:-------:|:-------------:|:---------:|:----------:|:-------:|:-------:|:---------:|:------:|:---------:|
|           |         |               |           |            |         |         |           |        |           |
| **CNN Dailymail dataset** | | | | | | | | | |
| **T5-search** | 58.12 | 0.11/7.726 | 0.24 | 6.69% | 7.66 | 34.48 | 0.66 | 45.51 | - |
| **T5-sample** | 67.58 | 0.11/7.790 | 0.20 | 3.50% | 5.05 | 30.15 | 0.64 | 48.51 | - |
| **Genie** | 179.9 | 0.09/7.293 | 0.24 | **4.16%** | 3.22 | 30.47 | 0.58 | 40.94 | 27.21 |
| **Genie$^{(10)}$** | 170.6 | 0.10/7.355 | 0.24 | 4.32% | 6.48 | **37.09** | 0.62 | 40.81 | - |
| **PLANNER** | 49.21 | **0.10/8.037** | 0.15 | 5.25% | 6.92 | 30.43 | 0.62 | 52.33 | **43.91** |
| **PLANNER$^{(10)}$** | 49.07 | 0.10/8.019 | **0.15** | 4.96% | **11.42** | 36.81 | **0.66** | 53.14 | - |
| **Human** | 49.477 | 0.12/8.226 | 0.16 | 5.63% | - | - | - | 51.15 | - |
|           |         |               |           |            |         |         |           |        |           |
| **XSum dataset** | | | | | | | | | |
| **T5-search** | 29.41 | 0.12/7.200 | 0.31 | 14.83% | 6.11 | 36.08 | 0.74 | 18.97 | - |
| **T5-sample** | 36.17 | 0.13/7.449 | 0.24 | 6.47% | 3.62 | 31.18 | 0.71 | 20.78 | - |
| **Genie** | 186.7 | 0.09/6.935 | 0.28 | 8.56% | 2.38 | 34.85 | 0.66 | 20.44 | 30.85 |
| **Genie$^{(10)}$** | 178.2 | 0.09/6.924 | 0.30 | 9.66% | 5.06 | **41.59** | 0.68 | 19.97 | - |
| **PLANNER** | 67.94 | **0.11/7.553** | **0.21** | **5.38%** | 4.84 | 33.97 | 0.69 | 20.04 | **57.88** |
| **PLANNER$^{(10)}$** | 67.46 | 0.11/7.529 | 0.23 | 5.82% | **11.61** | 41.23 | **0.72** | 19.89 | - |
| **Human** | 37.8 | 0.13/7.656 | 0.21 | 5.56% | - | - | - | 21.19 | - |

*2. Motivation and Impact of $k$*

The parameter $k$ determines the number of latent codes used to represent a paragraph and therefore controls the compression level. Latent codes with smaller values of $k$ are easier to model using the diffusion model, but may struggle to accurately preserve all the information in the original text. Additionally, smaller values of $k$ offer computational efficiency. as the sequence length for the diffusion model is $k$.

To determine the best set of latent codes, we indeed conducted experiments using three different methods: 1) selecting the first $k$ hidden vectors, 2) selecting the last $k$ hidden vectors, and 3) selecting interleaving hidden vectors, one for every $L/k$ hidden vectors. The results of the ablation study are presented below. Based on our findings, we observed no significant difference among the different choices, so we opted for option 1.

Furthermore, we discovered that increasing the value of $k$ does not lead to a dramatic improvement in performance (as stated on line 218, see the ablation study below). To tradeoff between efficiency and performance, in most of our study we focus on using $k=16$

| Experiment (on hotel review) | BLEU_clean | BLEU_robust |
|-----------------------------|------------|-------------|
| First k (k=16)              | 79.59      | 43.17       |
| Last k (k=16)               | 78.96      | 42.85       |
| Interleaving k (k=16)       | 79.81      | 43.35       |
| k=8                         | 57.90      | 30.68       |
| k=32                        | 82.31      | 45.14       |

---

### Decision · Program_Chairs · 2023-09-21

**Decision:**

Accept (poster)

**Comment:**

This paper introduces PLANNER, an innovative model that combines latent semantic diffusion with autoregressive generation to produce fluent text while maintaining global control over paragraphs. By integrating a "decoding" module with a "planning" module, the authors address issues of repetitive and low-quality output found in autoregressive models. The proposed method is evaluated on various conditional generation tasks, and the results demonstrate its effectiveness in generating high-quality long-form text efficiently. The reviewers are overall positive towards the contribution of this paper. I would recommend to accept this paper.